# The Impact of Small-Scale Greening on the Local Microclimate— A Case Study at Two School Buildings in Vienna

**Florian Teichmann** [1,*], **Andras Horvath** [2,*], **Markus Luisser** [2] **and Azra Korjenic** [1]

1    Institute of Material Technology, Building Physics and Building Ecology, TU Wien, 1040 Vienna, Austria
2    Rheologic GmbH, 1060 Vienna, Austria
*    Correspondence: florian.teichmann@tuwien.ac.at (F.T.); andras.horvath@rheologic.at (A.H.)

**Abstract:** Strategies to mitigate urban heat islands are a recent issue in the Austrian capital, Vienna. In this study, the uhiSolver-v2106-0.21 software was used to evaluate the summer cooling effects and humidity production of small-scale facade greening and a green pergola located in two schools within the city. Based on on-site measurement data, the study revealed that small-scale greening measures are not able to substantially reduce ambient air temperature. On a hot summer day, at 3 p.m. local time (CEST), the maximum decrease amounted to 0.3 °C at 0.1 m from the facade greening as well as inside the green pergola. As for the apparent (perceived) temperature, a reduction of up to 4 °C was observed under the green pergola compared to the unshaded roof terrace. Hence, the simulation results show that, within urban areas, a significant improvement of thermal comfort in summer can only be achieved through large-scale greenery that provides shade for pedestrians.

**Keywords:** building greenery; living walls; urban heat island (UHI) mitigation; urban microclimate; CFD simulation; apparent temperature





## 1. Introduction

Climate change, increasing urbanization, and re-densification in inner-city areas require new ways and solutions to increase the quality of life and living comfort in urban structures. Urban areas and agglomerations cause temperature increases due to the high percentage of sealed surfaces and the evaporation being reduced, while at the same time solar radiation is stored on building and street surfaces [1]. This leads to the formation of urban heat islands (UHI), where temperatures can sometimes be considerably higher than in surrounding areas. Summer heat leads to reduced quality of life and well-being of the population and can even lead to health consequences, especially for vulnerable population groups, such as old or sick people, as well as children. In the future—due to further increases in average temperatures and increased occurrence of extreme events, such as heat, floods, and drought—the urban system will need to gain resilience to ensure the quality of life of its inhabitants [2]. Against the background of climate change, the urban (micro-)climate is to be regarded as particularly sensitive and susceptible to disturbances. Due to the special spatial situation in cities with many sealed surfaces, a high density of building masses, and a strong heat production as a result of concentrated anthropogenic uses, it is necessary to pay special attention to measures that counteract UHI and other negative effects.

Proposed UHI mitigation strategies are cool pavements or roofs (with light-colored or permeable surfacing) and the increased use of vegetation [3]. While greening in the form of new urban parks is increasingly unlikely in a densely built city, buildings often provide large vacant areas which are extremely valuable for vertical and horizontal greening, with the area available for green walls being about twice that for green roofs [4]. The benefits of vertical greenery systems are covered extensively in the literature [5–11] and include mitigating the UHI effect [12], increasing outdoor thermal comfort [13], reducing noise [14],

eliminating pollutants [15], reducing building energy demand [16,17], increasing urban biodiversity [18], and other social and economic aspects [19].

### 1.1. Previous Research Projects on the Influence of Building Greenery Systems

The influence of vertical greenery systems on the microclimate within the inner courtyard of the BRG7 school in Kandlgasse, Vienna, was already investigated in the research project GrünPlusSchule (GreenPlusSchool). The cooling capacity of a living wall due to evapotranspiration, as well as the effect on the surface temperature of the outer facade due to shading by the vertical greenery system, were determined [20]. A computational fluid dynamics (CFD) simulation with the uhiSolver software was used to show the influence of the greening measures. These included two small-scale living walls, direct and indirect green facades, as well as a tree and some shrubs. The simulation results indicated that the influence on the air temperature in the inner courtyard of the school could be neglected [21]. Furthermore, it was found that due to the locally higher irradiance caused by reflections on the building facades, the lower air velocities, and the consequently higher air humidity, the apparent (perceived) temperature in the inner courtyard was in some locations higher than in the adjacent street canyons, despite the lower air temperature.

A recent study by Daemei et al. from 2021 investigated the thermal performance of a green facade compared to a bare wall on the north facade of a two-storey residential building in the humid climate of Rasht during summertime [22]. Using the ENVI-met software to evaluate the effect of the green facade on the surrounding air, it is stated that the temperature in front of the green facade is only insignificantly lower than in front of the bare wall. The peak temperate reduction within the ENVI-met simulation was 0.36 °C at 12 p.m.

Fahed et al. also used the ENVI-met software to conduct simulation studies on different urban heat island mitigation scenarios for the city of Beirut in Lebanon [23]. A reduction in air temperature during the day of up to 2 °C was observed when adding 7% of urban vegetation. An even greater reduction in local air temperature with a maximum of 7 °C was obtained at 1:00 p.m. and 1:00 a.m. by the blue scenario based on the implementation of fountains and water sprays. The third scenario, using high albedo materials, led to an increase in the mean radiant temperature, contributing to an additional thermal discomfort for pedestrians. Therefore, a combination of white models with other models is recommended to improve the pedestrian comfort.

Using computational fluid dynamics (CFD) simulations, a maximum temperature reduction of 2 °C within a street canyon in the city center of Arnhem, the Netherlands, was obtained by Gromke et al. when applying vegetation as avenue-trees, green facades, and roof greening at the same time [24]. Green facades only partially contributed to the cooling effect, with a maximum reduction of about 0.3 °C, whereas roof greening did not influence the air temperature inside the canyon at all. Therefore, the most significant reduction in temperature by a single greening measure was achieved with the avenue trees, reducing air temperature by a maximum of 1.6 °C at pedestrian level. In general, the cooling effect was limited to a distance of a few meters from the greening measures.

Another case study by Morakinyo et al. analyzed the thermal benefits of vertical greenery systems, such as living walls, in the high-density city of Hong Kong. Using the ENVI-met software combined with a parametric study, it was shown that 30 to 50% of all facades would need to be greened to potentially reduce air temperature by about 1 °C [25]. To maximize thermal comfort at the pedestrian level, vertical greenery systems should be placed at the base of buildings rather than at tower heights.

Similar results have been obtained by Peng et al. using the ENVI-met software to evaluate the cooling effects of block-scale facade greening with living walls in summer for the city of Nanjing in China [26]. For 30 scenarios with 6 idealized urban block forms and 5 different facade greening ratios, a maximum cooling intensity of 0.96 °C was determined for the high-rise, high-density scenario with a building height of 48 m, a building density of 34.6%, and 100% of the facades covered with living walls.

In the city of Colombo, Sri Lanka, the performance of vertical greenery systems in tropical conditions was evaluated by Galagoda et al. The study included three different types of green infrastructures, namely living walls, indirect green facades, and direct green facades [27]. Compared to a bare wall, the external surface temperature behind the vertical greenery systems was reduced by 8.72 K, 8.69 K, and 5.87 K for living walls, indirect green facades, and direct green facades, respectively. The air temperature, at a distance of 1 m from the vertical greenery systems, was reduced by 0.36 K, 0.61 K, and 0.01 K, respectively.

### 1.2. Assessing the Impact of Building Greenery Systems on the Local Microclimate

In the research project GRÜNEzukunftSCHULEN (GREENfutureSCHOOLS), two schools in Vienna (BRG16 Schuhmeierplatz and BRG15 Diefenbachgasse) were equipped with living wall systems in interior spaces and on north-facing exterior facades [28]. The living walls were accompanied by hygrothermal, microclimatic, and social measurements. The aim was to provide schools with green infrastructure and integrate it into the teaching and education of pupils. As part of the ongoing research project GRÜNEzukunftSCHULEN [2] (GREENfutureSCHOOLS [2]) the influence of the installed green infrastructure on the immediate microclimate in the outdoor areas is being investigated. The aim is to evaluate whether and how the greenery systems affect the sojourn quality in school open spaces. Unlike previous studies that took point measurements of outdoor temperature and air humidity, this project deploys a network of sensors set up on-site to draw conclusions about the spatial impact of greenery systems (two living wall systems and a green pergola).

The measurement data obtained is subsequently compared to a dynamic microclimate simulation using the uhiSolver software, version v2106-0.21 of Rheologic GmbH, Vienna. Compared to the ENVI-met software, which represents a microclimate simulation on a large scale with a horizontal resolution of a few meters [29], uhiSolver simulates the building and its immediate surroundings with a scale down to about 0.3 m. Compared to the ENVI-met simulations, the resolution is several times higher in certain areas. Furthermore, the cell geometry is not limited to right angles, so that deviating surfaces (e.g., roof slopes) and curves (columns, curved structures, etc.) can be mapped much more accurately. This is a particular advantage of the uhiSolver method when it comes to the correct mapping of near-wall flows in microclimate simulations. Hence, it is possible to realistically represent the complex relationships between solar radiation, air currents, the cooling capacity of plants through evaporation, and the heat storage of the surrounding surfaces. This provides the basis for a deeper understanding of the effect of greenery systems on the immediate microclimate close to the building. A detailed description of the functioning and model structure of uhiSolver can be obtained from the article "Simulation of urban microclimate with uhiSolver: software validation using simplified material data" [21]. Two additional validation cases of the Gablenzgasse (summer 2021) and the Naschmarkt area (summer 2022) in Vienna are provided in Appendix C, respectively. The present version of uhiSolver has an absolute deviation in terms of temperature and absolute humidity of 0.09 °C and 0.0042 $g_{water}/g_{air}$ and a standard deviation of $\pm1.13$ °C (3.57%) and $\pm0.0019$ $g_{water}/g_{air}$, respectively.

The microclimate simulations within the present study, examining small-scale greening measures at the schools BRG16 Schuhmeierplatz and BRG15 Diefenbachgasse in Vienna, are also using the simplified material data described in the aforementioned article. The research goal is to compare experimental data and CFD simulations of local outdoor effects on thermodynamic states (air temperature and absolute humidity), incident radiation, and apparent temperature (AT) of small-scale living walls and a green pergola with climbing vines.

## 2. Materials and Methods

Within this study, two living walls are investigated, one installed by Optigrün at Diefenbachgasse School, the other one built by Techmetall at Schuhmeierplatz School. The research also includes a green pergola at Schuhmeierplatz School as an example for a

low-tech greenery system. The measurements cover the analysis of air temperature and humidity, as well as wind speed, and are taken at different distances from the greenery systems. The measurement setup is described in Section 2.1. for Diefenbachgasse School and in Section 2.2. for Schuhmeierplatz School. The measurement data obtained directly from the experiments do not serve as a basis for the dynamic microclimate simulations with the uhiSolver software. Instead, external forcing (temperature, humidity, wind-speed) is applied using weather data from publicly available data of a weather station in Vienna's third district. The advantages of simultaneously collecting on-site measurement data and processing an independent microclimate simulation are that the measured data can be used to validate the new software, and that the simulation enables a differentiated processing and evaluation of the effects of the greenery systems on the microclimate (air temperature, relative humidity, radiation density, air currents, perceived temperature, etc.).

For both schools, the assessment is made for an exemplary model day with a clear sky and high outdoor air temperatures. The microclimate simulations presented are limited to the timeframe between 06:20 a.m. to 4:00 p.m. local time (CEST). The visualizations of the influence of the greening measures on the air humidity, air temperature, and apparent (perceived) temperature are conducted at 3:00 p.m. local time (CEST), the hottest time of day, to show the maximum influence. Therefore, nighttime transpiration is not evaluated as, according to Dayer et al., its contribution to the water stress of plants is negligible [30], and no significant impacts of plant's transpiration on air humidity and air temperature are expected for this time of day. However, evaporation of water from the substrates continues throughout the night, depending on local wind conditions, soil saturation, etc.

### 2.1. School BRG15 Diefenbachgasse

At the school BRG15 Diefenbachgasse, the living wall on the roof terrace, third floor, is the subject of the investigation. The living wall has a height of 2.4 m, a width of 4.9 m, and is mounted on the exterior wall with an air gap of 5 cm. It is equipped with a water tank and a circulation pump. Irrigation is performed twice a day, with a water consumption of 1.22 L/m²d. The plants used are *Geranium macrorrhizum*, helianthemum species, phlox species, and heuchera species. A schematic vertical section of the living wall is given in Figure 1. The weather station was attached to a 2 m high stand in the center of the terrace. Six temperature and humidity sensors were connected with cords, extending between the stand and the living wall at a height of 1.6 m. A seventh sensor was placed in the center of the living wall about 0.3 m in front of the planting. All seven sensors were equipped with a radiation shield. In addition, one surface temperature sensor was mounted on the exterior wall directly above the living wall and one was mounted on the concrete slabs on the ground in front of the living wall. The thermocouples used in this study were Rotronic HC2As (material polycarbonate) with an accuracy of $\pm$0.8% RH and $\pm$0.1 K at 10 to 30 °C, recording data every 5 min. The weather station used was a Davis Instruments Vantage Pro2™ (not ventilated), recording the following data every 15 min (the respective accuracies are set in brackets): relative humidity ($\pm$2%), rainfall ($\pm$3%), rain rate ($\pm$5%), solar radiation ($\pm$5%), temperature ($\pm$0.3 °C), wind direction ($\pm$3°), and wind speed ($\pm$0.9 m/s or $\pm$5%). The data logger used was a Driesen + Kern DataCollectorXP-R, with the measurement data being exported once a week. The positions of the sensors are illustrated schematically in Figure 2; the way they were mounted on-site is shown in Figure 3.

From the measurement data collected in summer 2021 at the weather station on the roof terrace of Diefenbachgasse School, a representative model day is selected for the subsequent microclimate simulation, which should represent a summer day with clear skies, peak outdoor air temperatures, and high solar irradiation. The selected day is July 28, which has a slightly lower solar irradiance (see Figure 4), but the highest outdoor air temperature of the entire series of measurements (see Figure 5). The local climate during the study period at the Diefenbachgasse School weather station is displayed in Table 1, which shows average, maximum, and minimum values of air temperature, absolute air humidity, wind velocity, and wind direction.

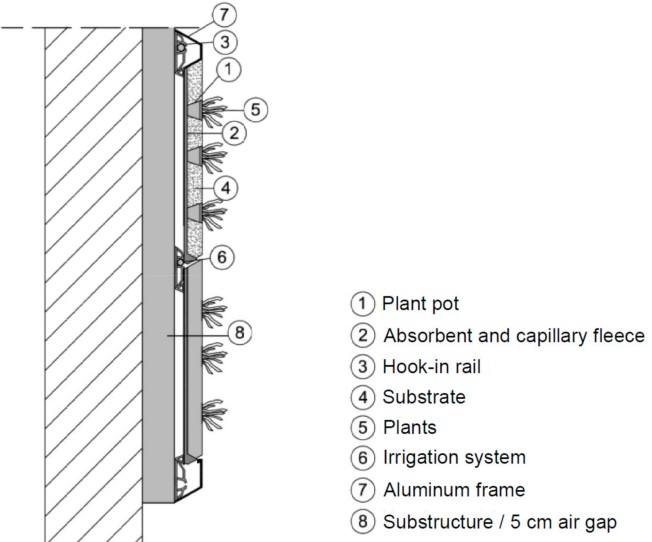

**Figure 1.** Schematic vertical section [31] of the living wall (cassette system) at Diefenbachgasse School, with the main elements given.

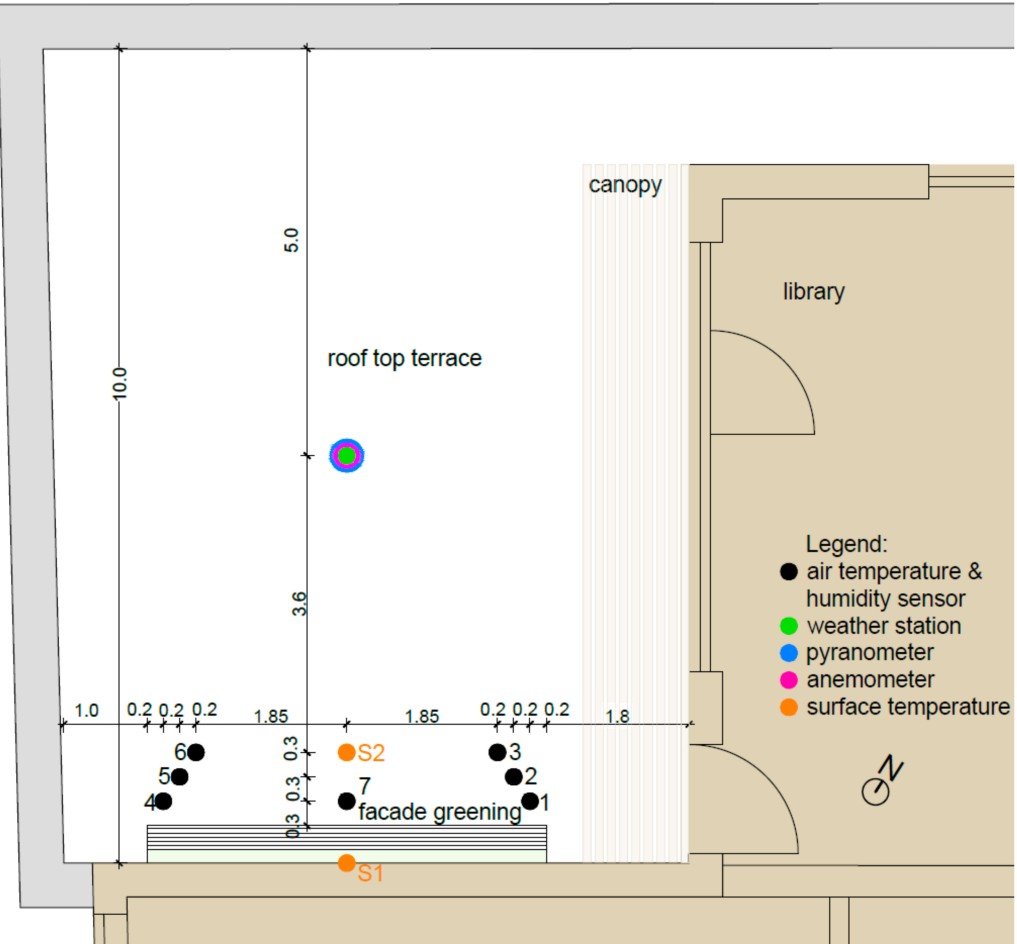

**Figure 2.** Schematic representation of the positions of the installed sensors at Diefenbachgasse School with the air temperature and humidity sensors 1 to 7 and the surface temperature sensors S1 and S2.

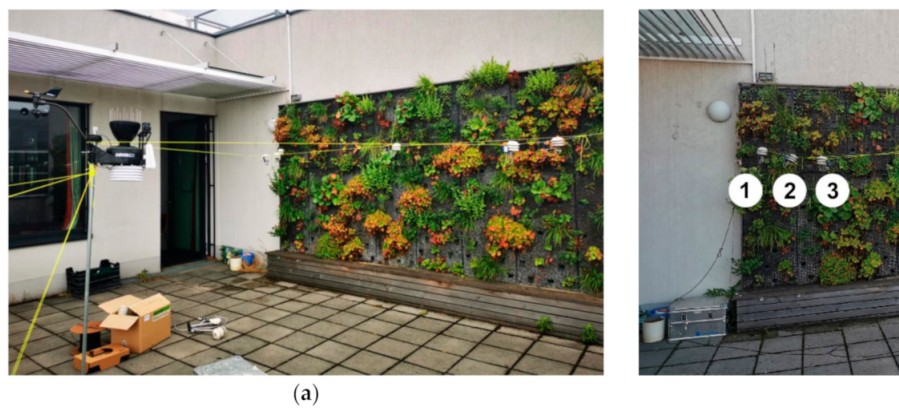
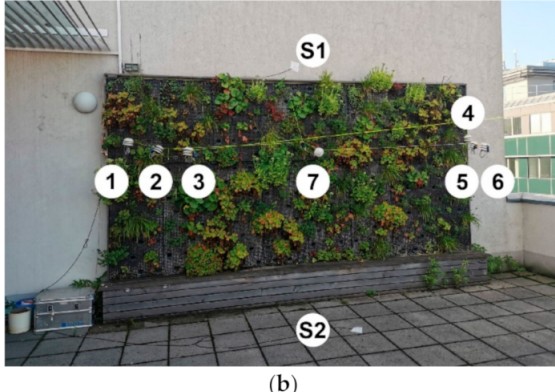

(**a**)  (**b**)

**Figure 3.** Images of the installation of the sensors at Diefenbachgasse School: (**a**) position of the weather station; (**b**) position of the temperature and humidity as well as surface temperature sensors.

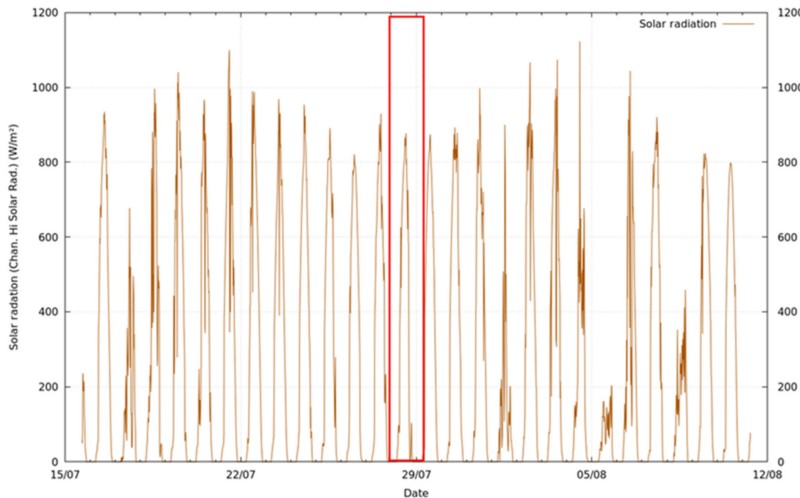

**Figure 4.** Solar radiation measured by the weather station at Diefenbachgasse School during the entire measurement period in summer 2021, with the selected model day highlighted.

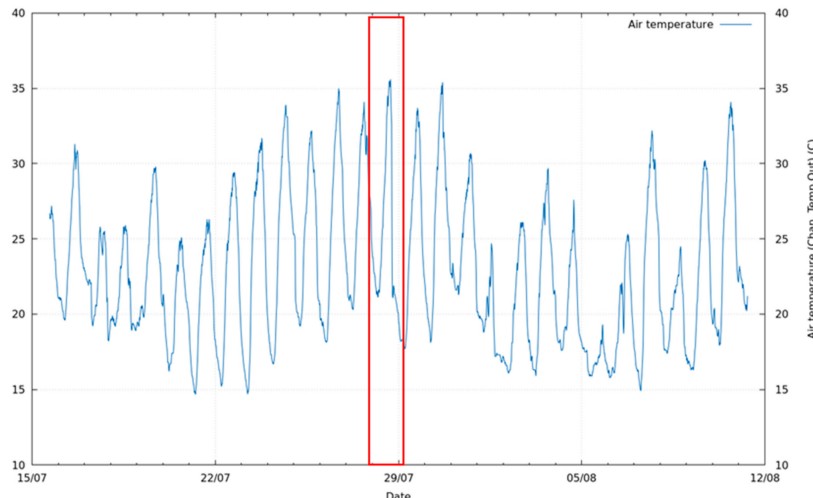

**Figure 5.** Outdoor air temperature measured by the weather station at Diefenbachgasse School during the entire measurement period in summer 2021, with the selected model day highlighted.

**Table 1.** Average, maximum, and minimum values of air temperature [T in °C], absolute humidity [X in kg water vapor/kg dry air], wind velocity [U in m/s], wind direction, and total solar irradiance [TSI in W/m$^2$] at the weather station of Diefenbachgasse School on the model day between 6:20 and 16:00 CEST.

|  | T (°C) | X (kg/kg) | U (m/s) | Wind-Dir. (-) | TSI (W/m$^2$) |
|---|---|---|---|---|---|
| Minimum | 21 | 0.0125 | 1.6 | ESE | 25 |
| Average | 28 | 0.0130 | 3.2 | W | 436 |
| Maximum | 35 | 0.0135 | 4.8 | WSW | 846 |

*2.2. School BRG16 Schuhmeierplatz*

At BRG16 Schuhmeierplatz, both the living wall in the inner courtyard and the green pergola on the roof terrace, third floor, were investigated. The L-shaped living wall is mounted on the exterior wall with an air gap of 5 cm. The height varies between 1.75 m and 3 m, and the width varies between 2 m and 8 m. The plants used are *Geranium macrorrhizum*, helianthemum species, and heuchera species. Irrigation is performed twice a day, with a water consumption of 1.22 L/m$^2$d. The green pergola covers an area of 3 m by 7 m and is 2.7 m high. The plants used are *Rubus fruticosus*, *Actinidia arguta*, fragaria, *Akebia uinata*, *Thunbergia alata*, *Santolina chamaecyparissus*, *Salvia nemorosa*, *Lavandula angustifolia*, helianthemum, coreopsis, *Tropaeolum majus*, and lonicera species. Irrigation is performed once a day, with a water consumption of 73.5 L/d, and it is carried out via a direct connection to the water pipe and regulated by means of an irrigation computer. A schematic vertical section of the living wall and the green pergola are given in Figure 6. Six temperature and humidity sensors were mounted at different positions in front of the living wall. In addition, three surface temperature sensors were affixed on the exterior wall approximately 2 m above the green façade (see Figure 7). The four temperature and humidity sensors in the pergola were attached to the wooden structure of the pergola in two different positions and at two different heights (see Figure 8). The weather station was again mounted on a 3 m high stand, next to the pergola. In addition, two surface temperature sensors were fixed on the concrete slab of the roof terrace. The thermocouples used at Schuhmeierplatz School were also Rotronic HC2As, recording data every 5 min. The weather station used was a Davis Instruments Vantage Pro2™ Aktiv, recording the same data as at Diefenbach School every 15 min (with the accuracies also being the same). The data logger used was a Keysight DAQ970A, with the measurement data being exported once a week. An overview of the positions of all sensors is provided in Figure 9. All temperature and humidity sensors were equipped with a radiation shield.

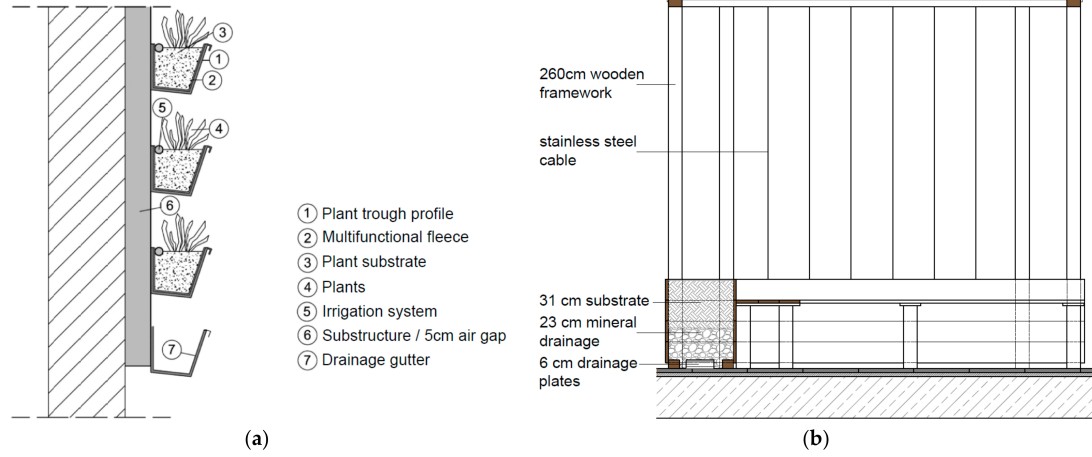

**Figure 6.** Schematic vertical sections of the greenery systems at Schuhmeierplatz School, with their main elements given: (**a**) living wall (trough system) [31]; (**b**) green pergola.

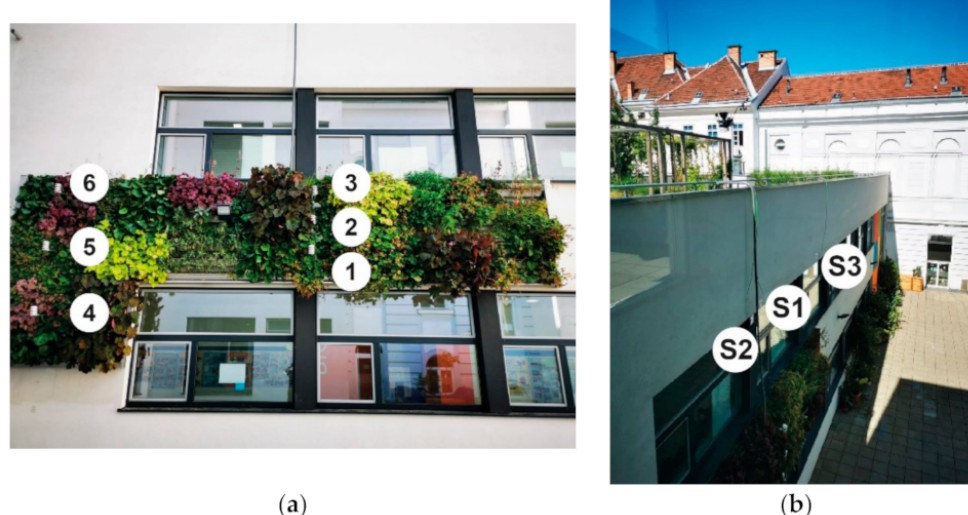

**Figure 7.** Images of the installed sensors at Schuhmeierplatz School: (**a**) position of the temperature and humidity sensors in front of the living wall; (**b**) position of the surface temperature sensors above the living wall, and position of the weather station on the roof terrace.

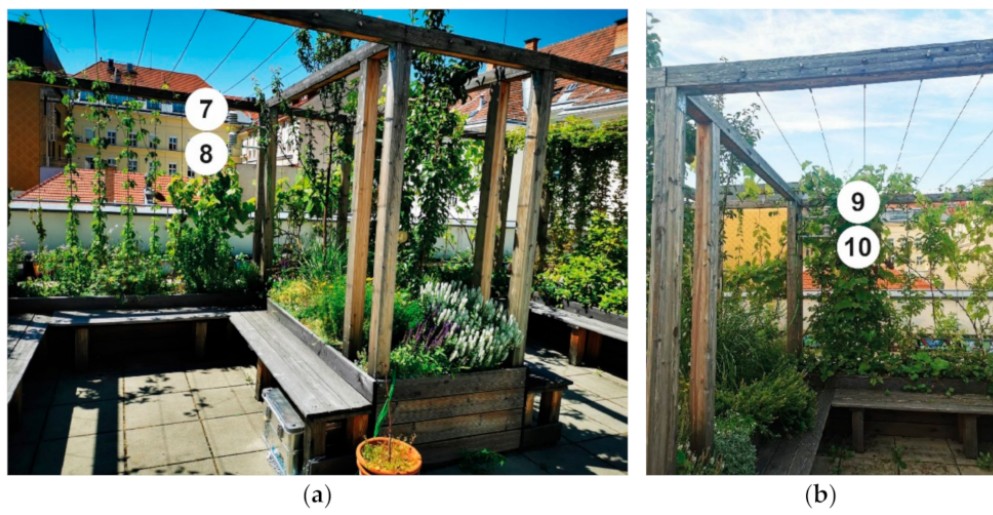

**Figure 8.** Images of the installed sensors at the green pergola at Schuhmeierplatz School: (**a**) position of the temperature and humidity sensors on the left side of the pergola; (**b**) position of the temperature and humidity sensors on the right side of the pergola.

Due to technical problems with the weather station on the roof terrace of Schuhmeierplatz school, the recorded measurement data are limited to the period of one week in August 2021. As with Diefenbachgasse School, a representative model day with the highest possible outdoor air temperature and consistently high solar radiation was chosen. The selected day is August 14, a day with a peak temperature of 35 °C and a nearly undisturbed solar radiation curve (see Figure 10). The local climate in the examined period at Schuhmeierplatz weather station is displayed in Table 2, showing average, maximum, and minimum values of air temperature, absolute air humidity, wind velocity, and wind direction.

Subsequently, wind simulations and microclimate simulations were carried out for the selected model days of the two schools using the uhiSolver software. For this purpose, weather data from a weather station in the third district of Vienna were used as input data in order to compare the results of the simulation with the measured data collected on-site.

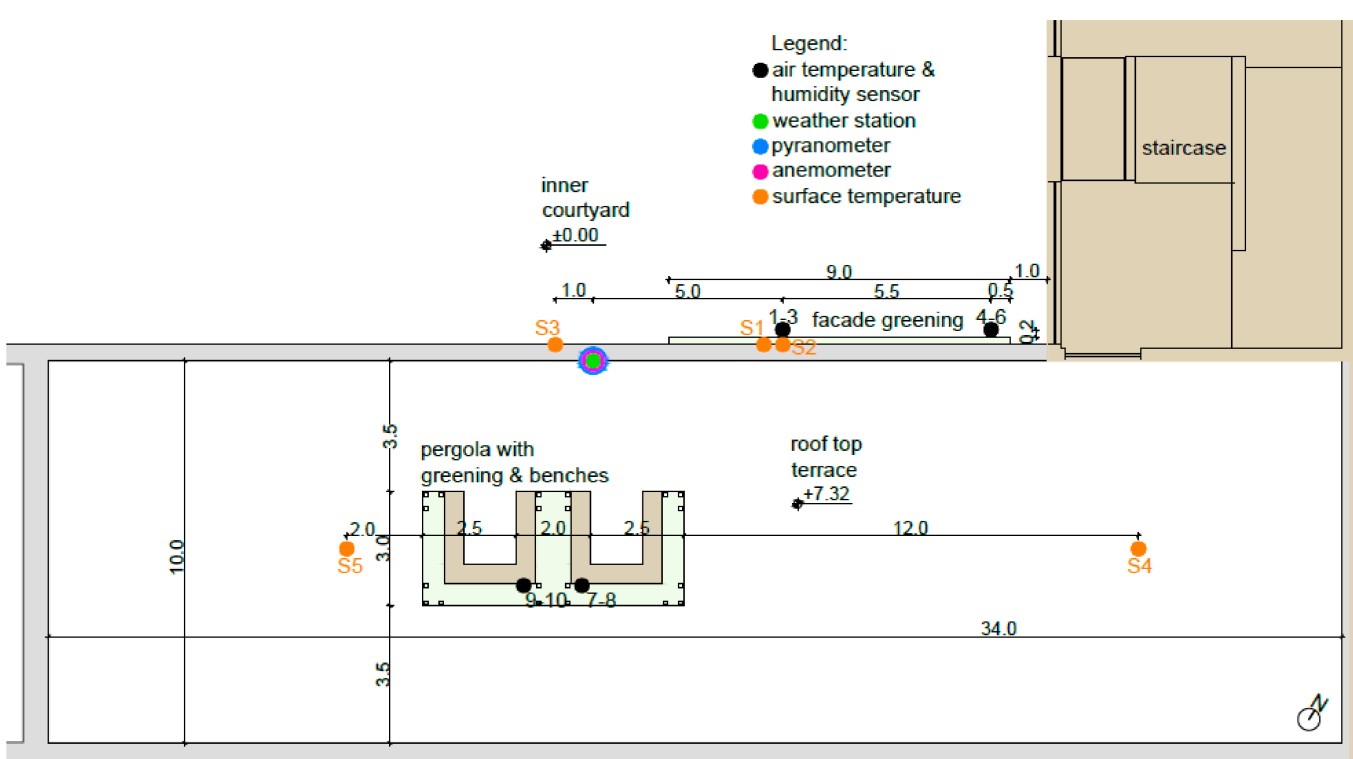

**Figure 9.** Schematic representation of the positions of the installed sensors at Schuhmeierplatz School with the air temperature and humidity sensors 1 to 10 and the surface temperature sensors S1 to S5.

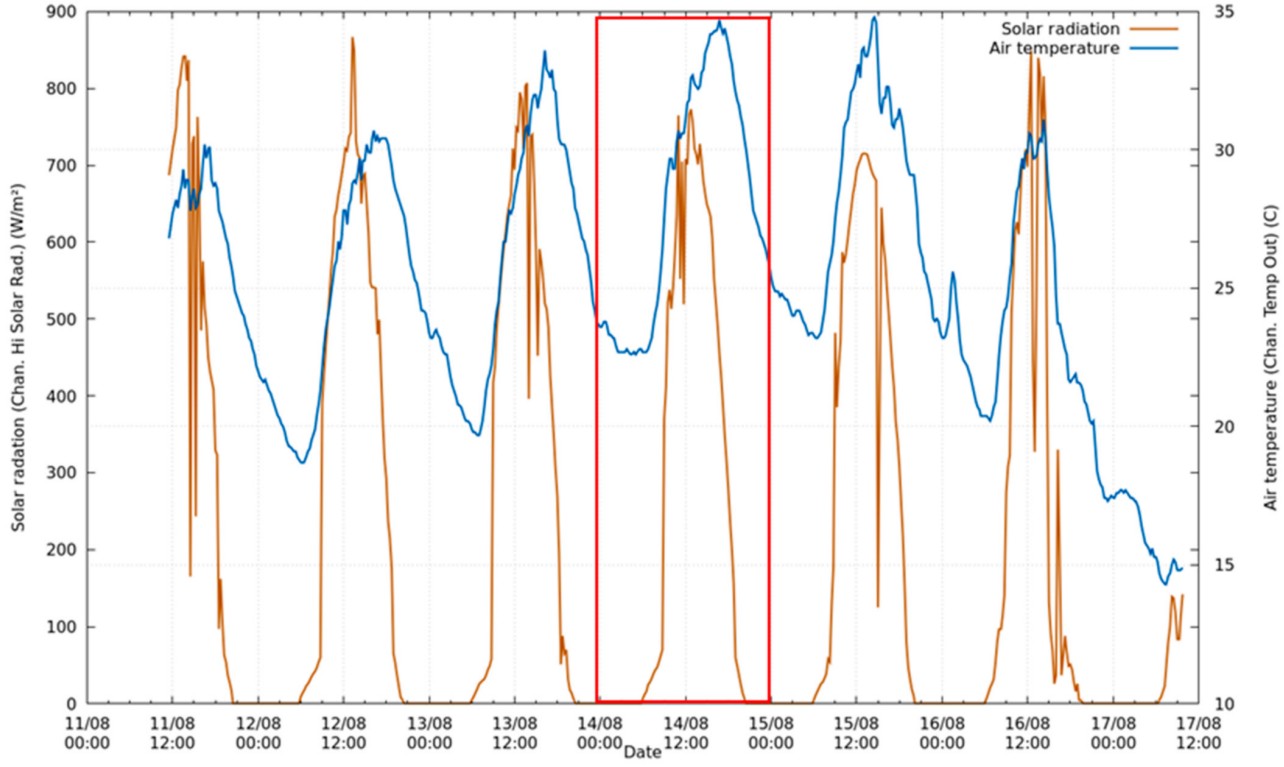

**Figure 10.** Outdoor air temperature and solar radiation measured by the weather station at Schuhmeierplatz School in August 2021, with the selected model day highlighted.

**Table 2.** Average, maximum, and minimum values of air temperature [T in °C], absolute humidity [X in kg water vapor/kg dry air], wind velocity [U in m/s], wind direction, and total solar irradiance [TSI in W/m²] at the weather station of Schuhmeierplatz School on the model day between 6:20 and 16:00 CEST.

|  | **T (°C)** | **X (kg/kg)** | **U (m/s)** | **Wind-Dir. (-)** | **TSI (W/m²)** |
|---|---|---|---|---|---|
| Minimum | 21.5 | 0.0120 | 0 | E | 0 |
| Average | 28.4 | 0.0128 | 2.4 | ESE | 367 |
| Maximum | 35.3 | 0.0135 | 4.8 | WSW | 735 |

## 3. Results

### 3.1. School BRG15 Diefenbachgasse

At the beginning, a wind simulation of the area of Diefenbachgasse School was carried out to determine the large-scale wind direction corresponding to the wind conditions on the roof terrace according to the measured data of the weather station. The Y coordinate (yellow arrow) of the coordinate system in Figure 11 points to the north, and the X coordinate (red arrow) to the east. A large-scale wind direction of WSW coincides best with the measured data from the weather station and, thus, was consequently used as a basis for the microclimate simulation. Figure 11b also shows that a wind vortex is formed on the roof terrace. Nevertheless, the wind direction at the weather station corresponds to that in front of the living wall. Thus, the measurement data of the weather station regarding the wind direction can be used for the determination of the wind conditions directly in front of the living wall. As the main large-scale wind direction in Vienna from July to September in the years 2009 to 2019 is west (see Figure 12), the wind conditions of the selected model day with a large-scale wind direction of WSW can be considered representative for the examined period.

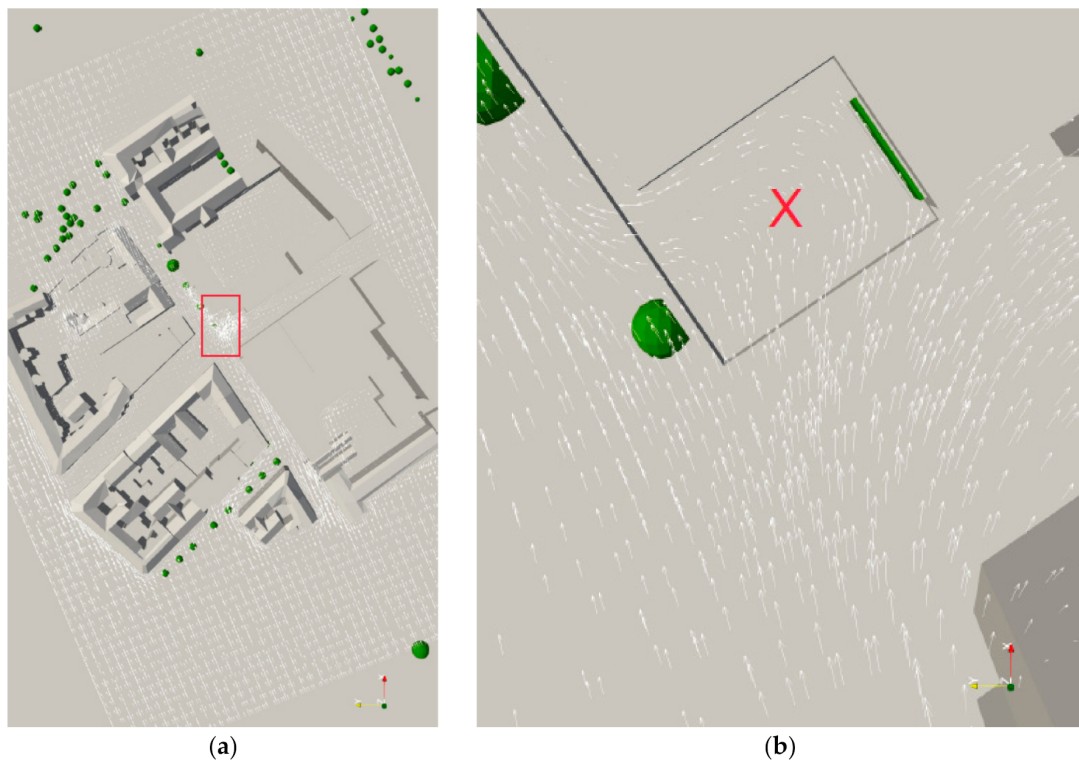

(**a**)  (**b**)

**Figure 11.** Results of the wind simulation at Diefenbachgasse School: (**a**) representation of the entire simulation area with the roof terrace highlighted; (**b**) wind conditions and position of the weather station on the roof terrace.

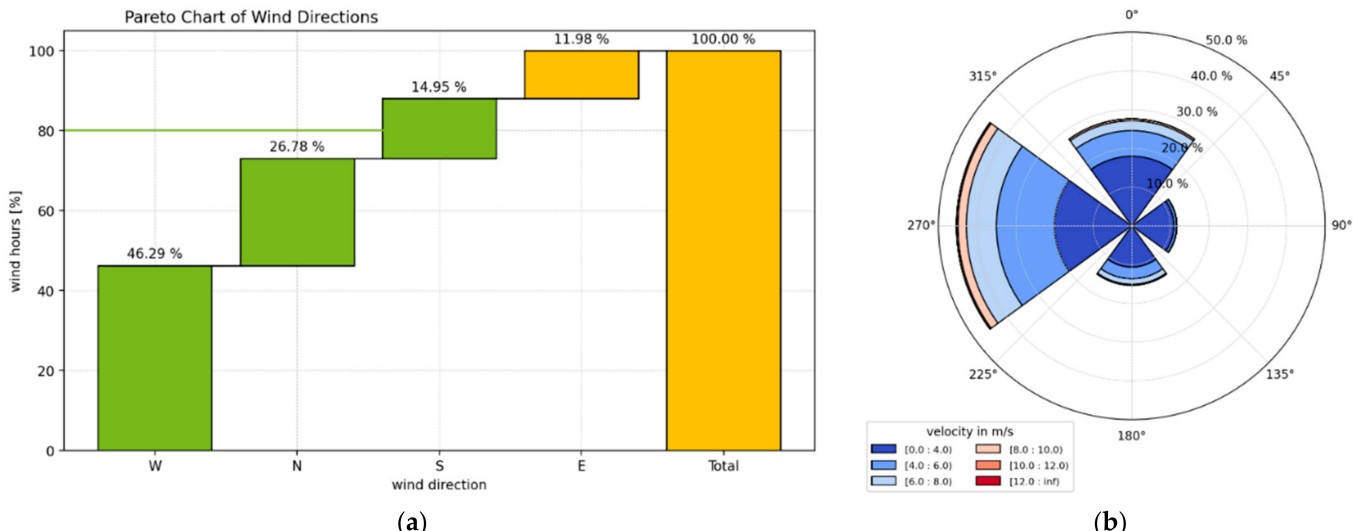

(**a**)                       (**b**)

**Figure 12.** Typical wind conditions in summer (July to September) in Vienna using hourly weather data from 26.8.2009 to 26.8.2019 [32]: (**a**) pareto chart of wind directions; (**b**) wind rose diagram of wind velocities in each wind direction.

Based on the results of the wind simulation, the next step was to perform a microclimate simulation for the object area. The simulation results are shown for the hottest time of the day at 3:00 p.m. CEST. The living wall faces north-west and is just about to move out of the shadow of the school building, whereas the roof terrace itself is directly exposed to the sun (see Figure 13).

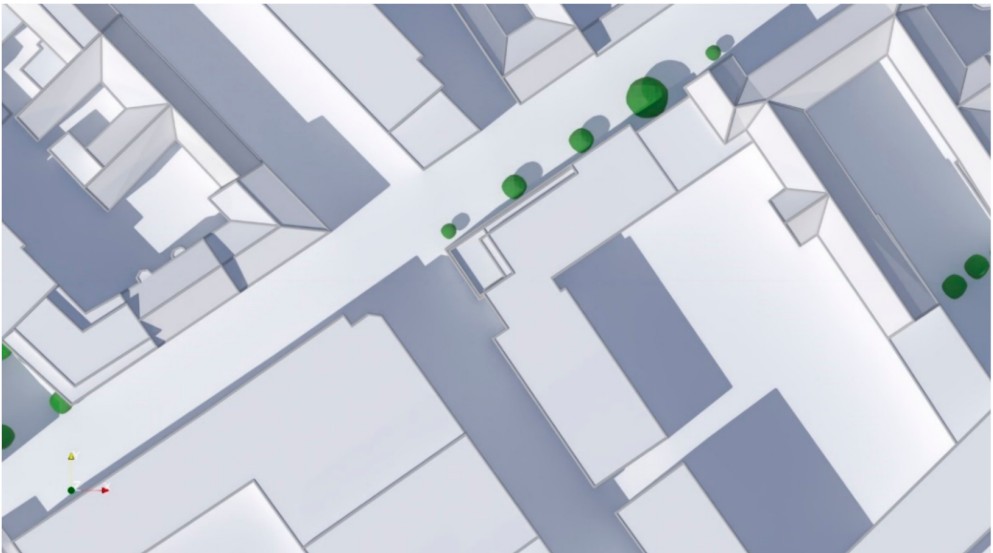

**Figure 13.** Shadow cast at Diefenbachgasse School on the model day at 3:00 p.m. CEST, with the roof terrace highlighted.

The results of the microclimate simulation are displayed as colored plots, with each color representing a different value according to the given color table. The evaluations of the humidity load (Figure 14) and the air temperature (Figure 15) on the model day at 3:00 p.m. CEST show that the effect of the living wall on the microclimate is limited to the immediate area in front of the greenery and downwind of the greenery. The moisture produced by the living wall is immediately removed by the wind. The cooling effect due to evapotranspiration of the plants can also be seen directly in front of the facade and

downwind of the living wall. The cool air is then carried away along the facade, with the cooling effect of the greenery gradually blending into that of the shading near the facade as the distance from the greenery increases. The contribution of the living wall to the air temperature reduction on the model day at 3:00 p.m. CEST is limited to 0.3 °C at a 0.1 m distance and 0.1 °C at a 0.5 m distance from the greening, respectively. At a distance of 1 m from the living wall, no influence on the air temperature can be detected.

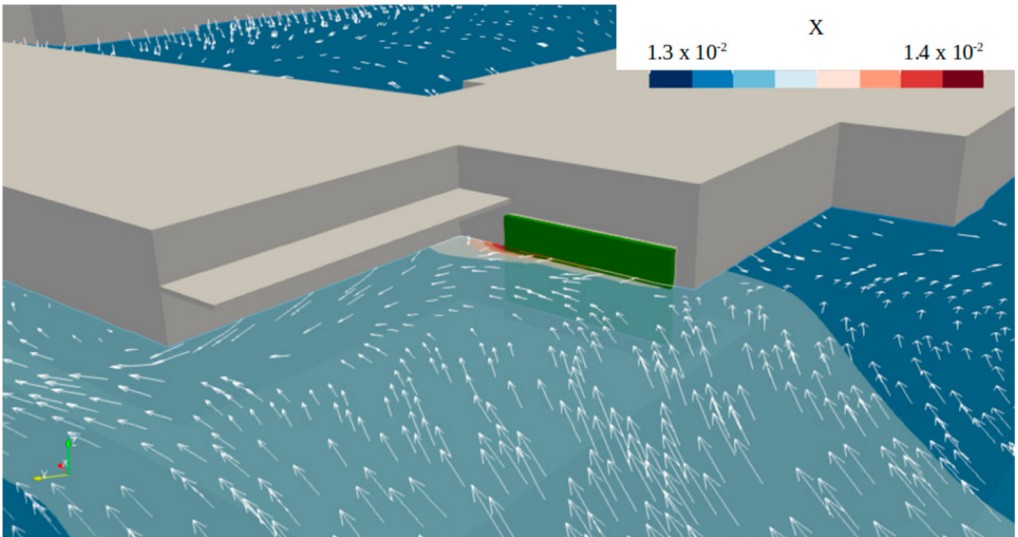

**Figure 14.** Humidity load (kg/kg) at the level of the Diefenbachgasse School weather station on the model day at 3:00 p.m. CEST.

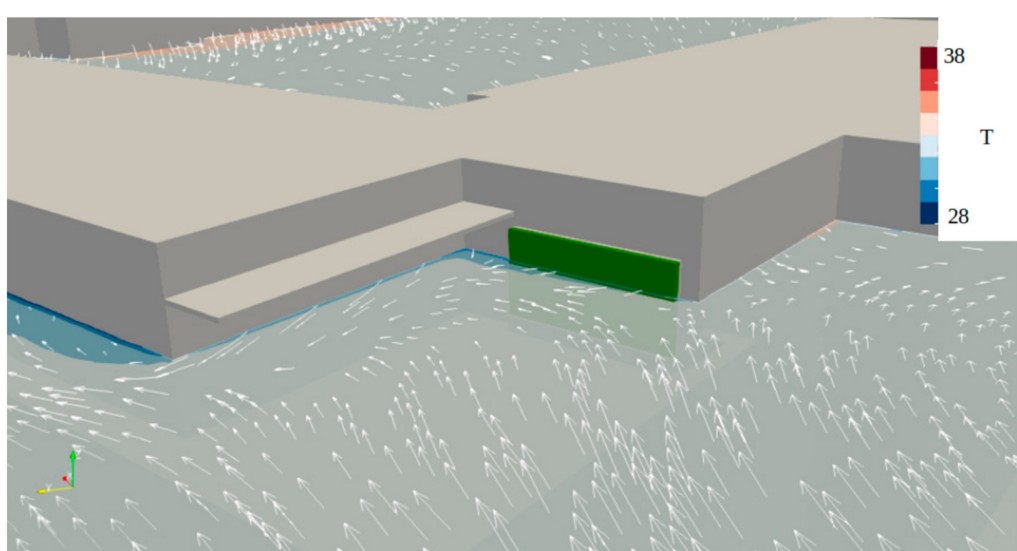

**Figure 15.** Air temperature (°C) at the level of the Diefenbachgasse School weather station on the model day at 3:00 p.m. CEST.

Observing the incident radiation intensity (Figure 16) and the surface temperatures (Figure 17) on the model day at 3:00 p.m. CEST, the following figures clearly show that there is considerable radiation concentration and high surface temperature in some areas. The north-west-oriented living wall can slightly change this unfavorable situation, which is due to the geometry and orientation of the building. It can only be observed that there are no such radiation and surface temperature peaks in the area in front of the living wall.

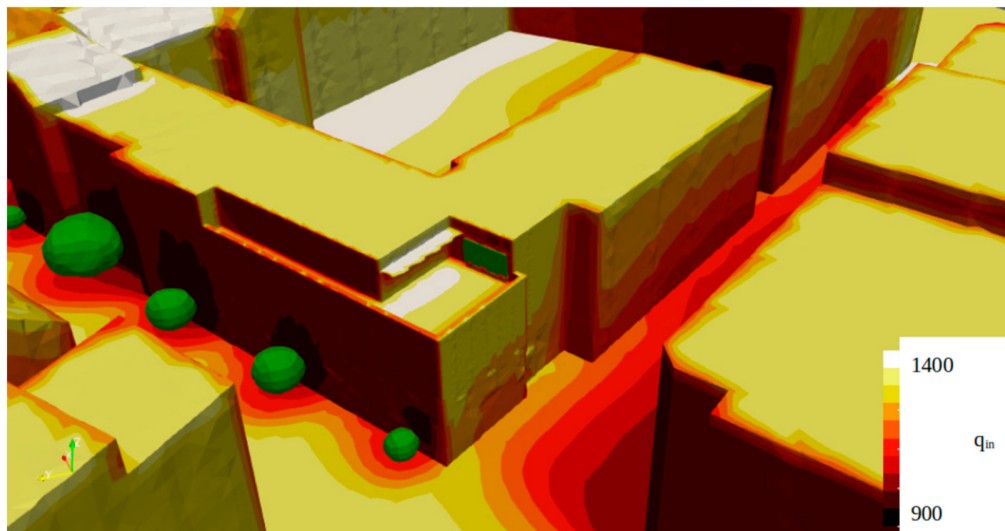

**Figure 16.** Incident surface radiation intensity (W/m$^2$) at Diefenbachgasse School on the model day at 3:00 p.m. CEST.

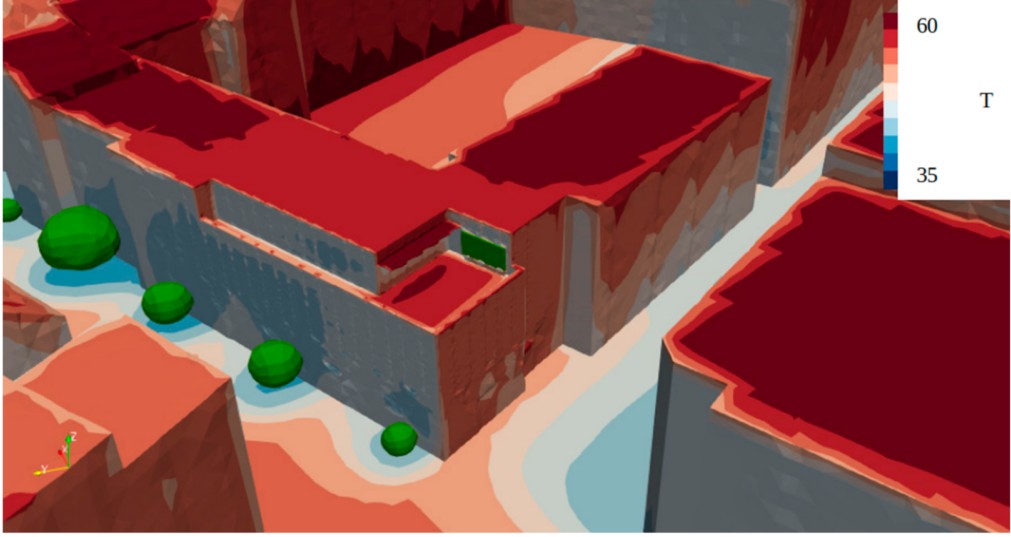

**Figure 17.** Surface temperatures (°C) at Diefenbachgasse School on the model day at 3:00 p.m. CEST.

The results of the simulation validation by means of the experimental data show an increasing deviation between measured and simulated values in the afternoon hours (see Figure 18) with an average absolute error of −2.78 °C and a standard deviation of absolute errors of 1.12 °C at 3:00 p.m. CEST. This could be due to the use of non-ventilated radiation shields. In the afternoon hours, the air temperature sensors are directly exposed to solar radiation, so the recorded air temperature might be influenced despite the attached radiation shields. Future measurements will use ventilated radiation shields to exclude the influence of solar irradiation on the measurement data. Further information on the validation results is provided in Appendix C.

In the following section, the results of the wind and microclimate simulation of Schuhmeierplatz School are presented.

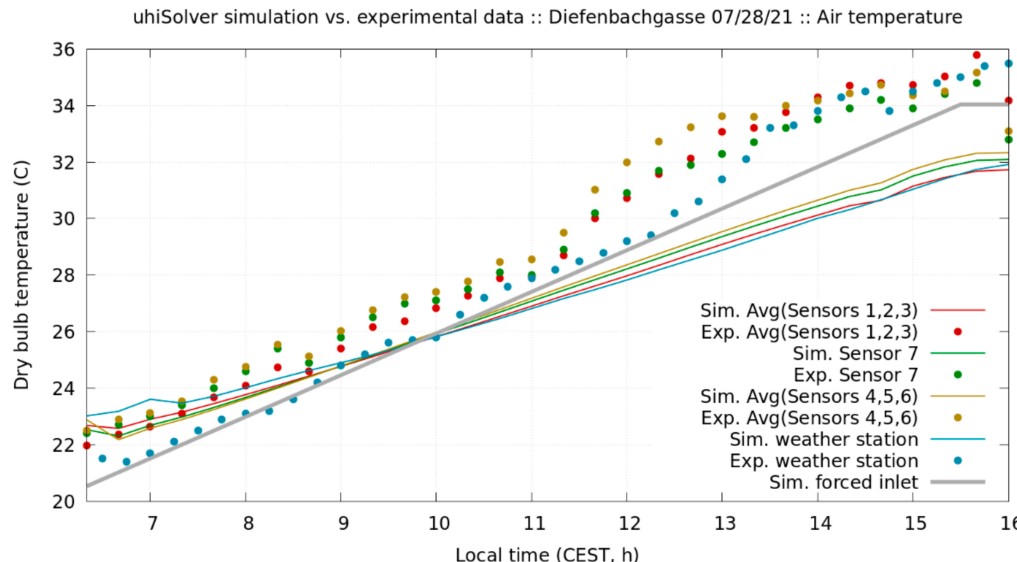

**Figure 18.** Validation of simulated air temperature with measured data at Diefenbachgasse School on the model day from 06:20 a.m. to 4:00 p.m. CEST.

### 3.2. School BRG16 Schuhmeierplatz

Several wind simulations were performed at Schuhmeierplatz School in advance to determine the large-scale wind direction in which the resulting wind conditions on the school's roof terrace would match the wind measurement data from the weather station. Of the four wind simulations performed with different large-scale wind directions, the one with westerly wind corresponds most closely to the locally measured wind direction E to ESE at the weather station (see Figure 19). This 180° deviation from the main wind direction is due to the formation of an air vortex at the examined building block. Again, the large-scale wind direction (west) on the selected model day corresponds to the main wind direction of the examined summer period in Vienna (see Figure 12).

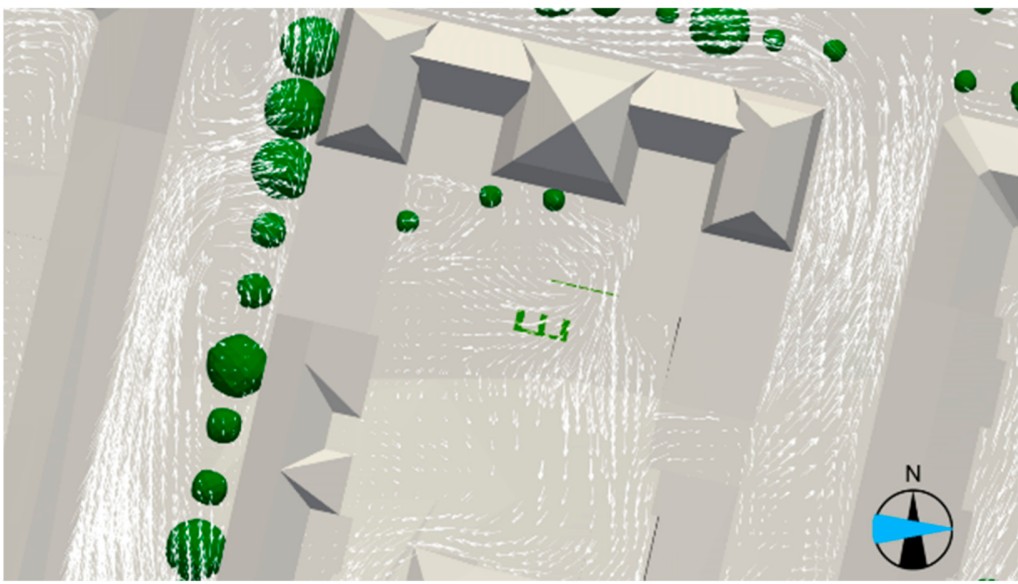

**Figure 19.** Result of the wind simulation at Schuhmeierplatz School for large-scale westward wind direction, as indicated by the blue arrow.

The simulation results are shown for the hottest time of the day at 3:00 p.m. CEST. The living wall within the inner courtyard faces north and, thus, is in the shadow of the

school building, while the green pergola on the roof terrace is directly exposed to the sun, providing shade only in a very confined area (see Figure 20).

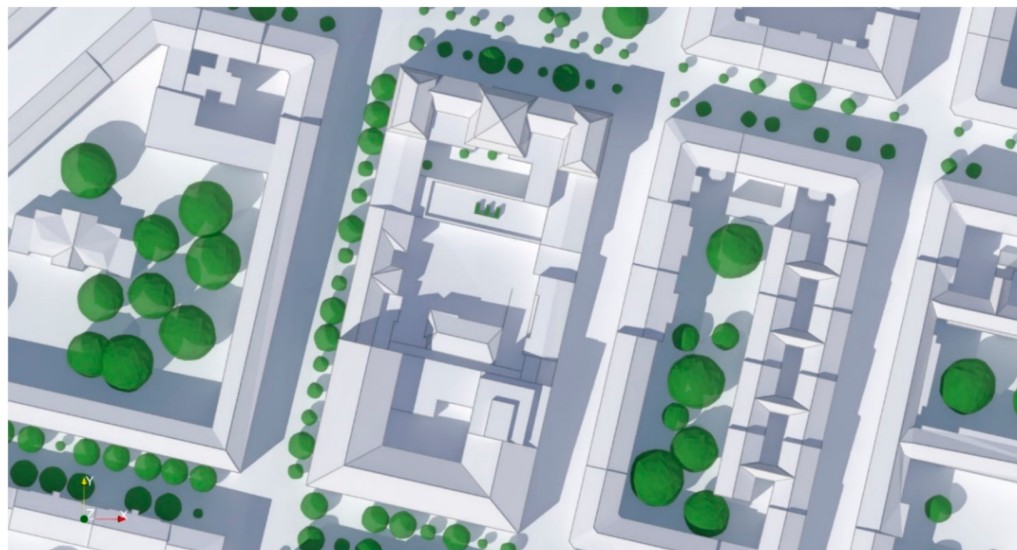

**Figure 20.** Shadow cast at Schuhmeierplatz School on the model day at 3:00 p.m. CEST, with the position of the greenery systems highlighted.

The microclimate simulation with uhiSolver using the large-scale westward wind direction shows a similar situation in the inner courtyard of Schuhmeierplatz School as on the roof terrace of Diefenbachgasse School, in that the humidity produced by the plants of the living wall is directly carried away by the wind (see Figure 21), resulting in a very limited spatial cooling effect. The increased humidity is, thus, only visible in the immediate vicinity in front of and a few meters downwind of the living wall. With respect to the thermodynamic air temperature in the inner courtyard on the model day at 3:00 p.m. CEST, within the numerical range of variation, no effect of the living wall is detectable in the CFD simulation (see Figure 22). The increased air temperature observed on the short side of the courtyard is due to downward winds along the heated facade.

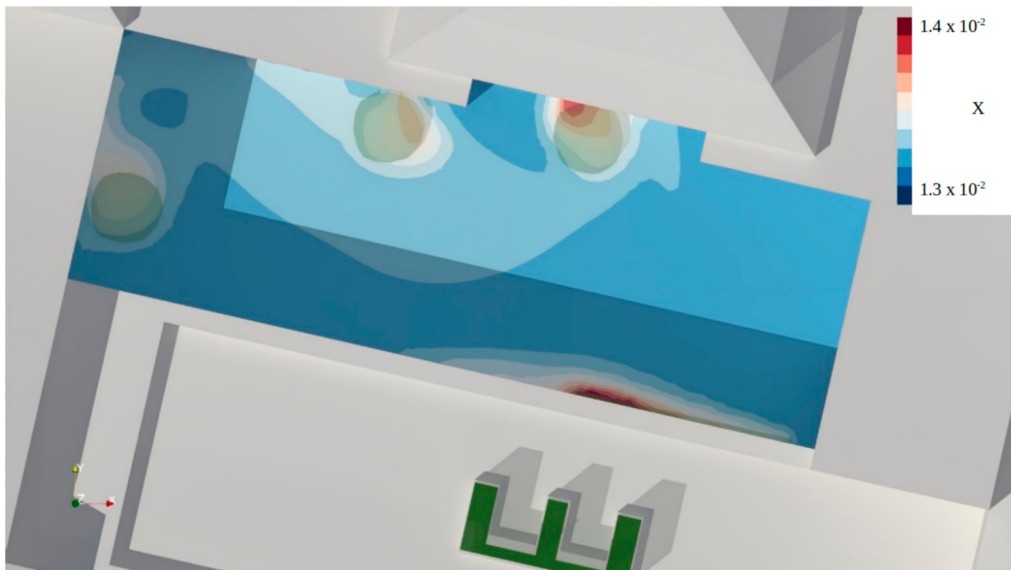

**Figure 21.** Humidity load (kg/kg) at the level of the Schuhmeierplatz School living wall on the model day at 3:00 p.m. CEST.

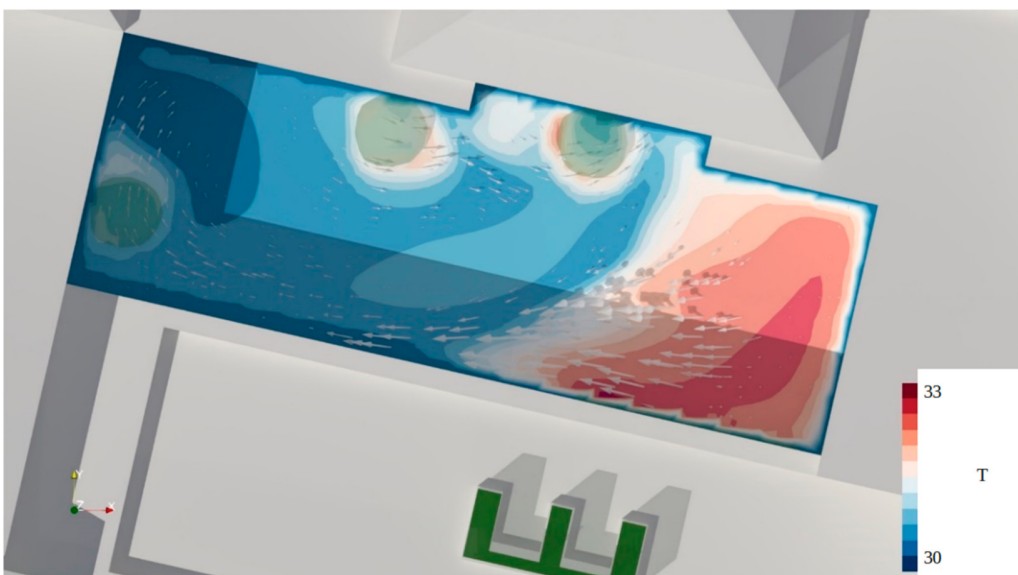

**Figure 22.** Air temperature (°C) at the level of the Schuhmeierplatz School living wall on the model day at 3:00 p.m. CEST.

Regarding the green pergola on the roof terrace, similar conclusions can be obtained. Whereas the impact of the plants on the absolute air humidity in the vicinity of the green pergola is significant (Figure 23), the air temperature is in some parts even higher than in the adjacent parts of the roof terrace (Figure 24). This can be attributed to the warming of the leaves themselves and the additional reflections of solar irradiance. The maximum air temperature reduction within the green pergola on the model day at 3:00 p.m. CEST is 0.3 °C, decreasing to 0.1 °C at about 1 to 3 m distance from the pergola. At a 5 m distance, there is no detectable impact of the green pergola on air temperature.

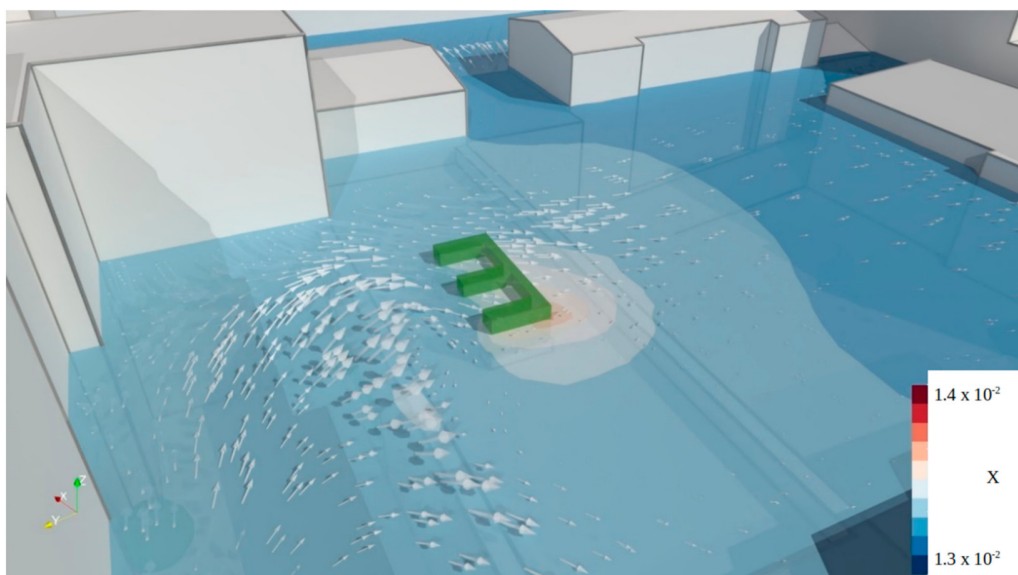

**Figure 23.** Humidity load (kg/kg) at the level of the Schuhmeierplatz School green pergola, 1.6 m above the ground, on the model day at 3:00 p.m. CEST.

In terms of the apparent (perceived) temperature by Steadman [33], a reduction of about 4 °C can be noticed within the shade of the green pergola compared to the unshaded parts of the terrace, as shown in Figure 25. However, the cooling effect is primarily

dominated by the shading effect and less by the evapotranspiration performance of the plants.

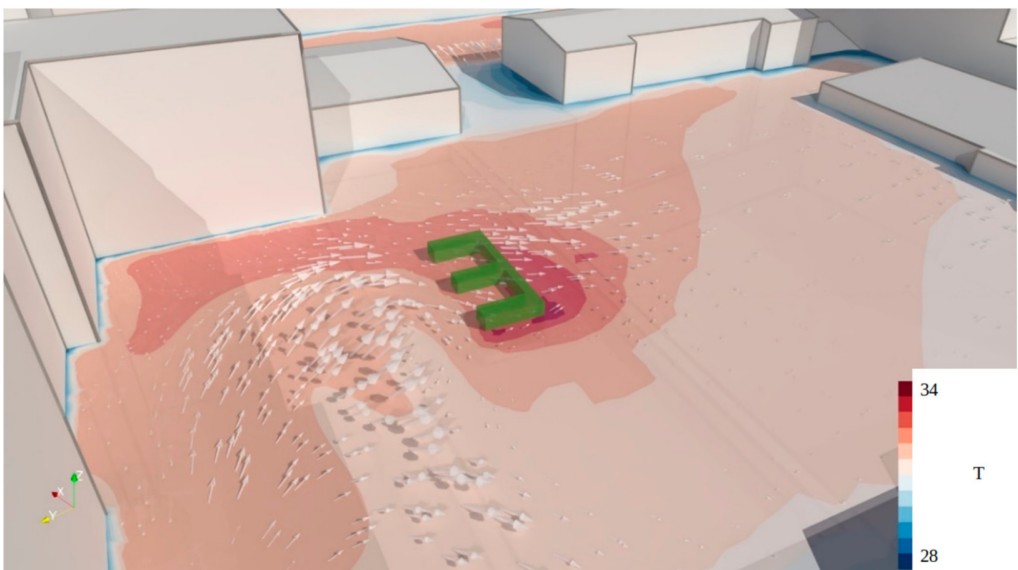

**Figure 24.** Air temperature (°C) at the level of the Schuhmeierplatz School green pergola, 1.6 m above the ground, on the model day at 3:00 p.m. CEST.

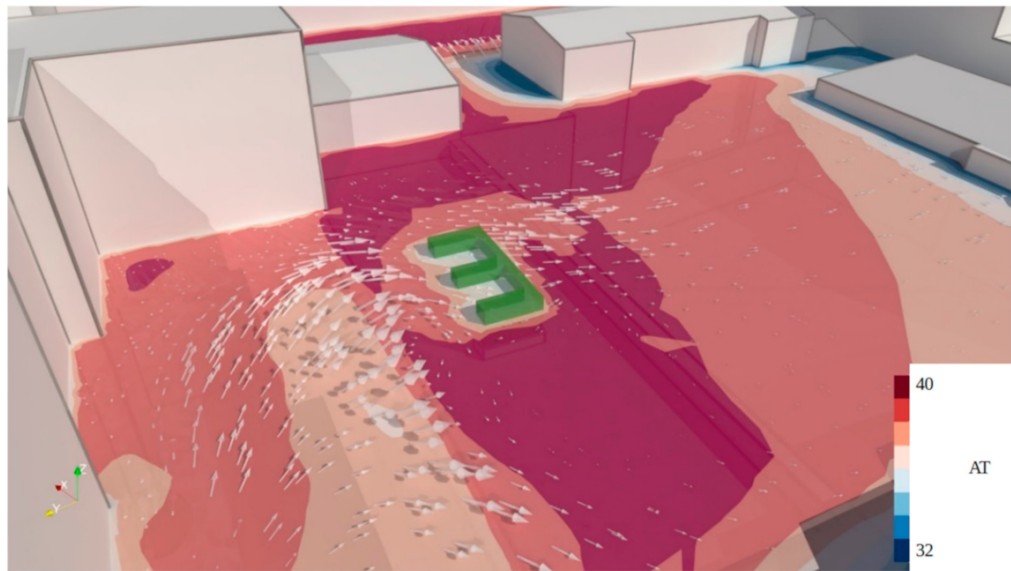

**Figure 25.** Apparent temperature (°C) at pedestrian level on the roof terrace of Schuhmeierplatz School on the model day at 3:00 p.m. CEST.

As indicated in the plots of apparent temperature (Figure 26), surface temperatures (Figure 27), and incident radiation intensity (Figure 28) on the model day at 3:00 p.m. CEST, the more exposed roof surfaces and roof terraces are heated significantly more than shaded areas, such as the school's courtyard or the school forecourt. High incident radiation intensities and, consequently, high surface temperatures can also be seen on the roof terrace with the green pergola. Due to its small size, the reduction in the apparent temperature is limited to the space underneath. From a large-scale perspective, the influence of the green pergola seems to be negligible.

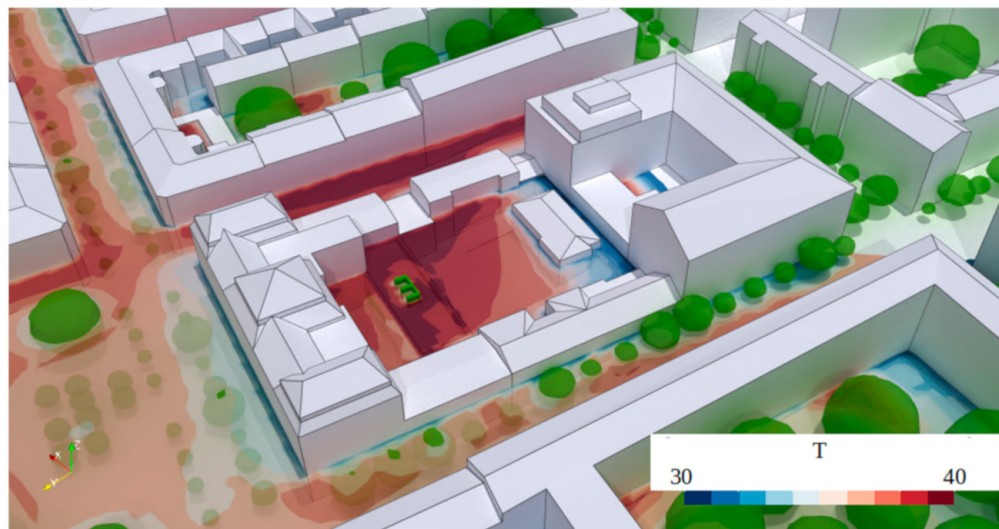

**Figure 26.** Apparent temperature (°C) around Schuhmeierplatz School on the model day at 3:00 p.m. CEST.

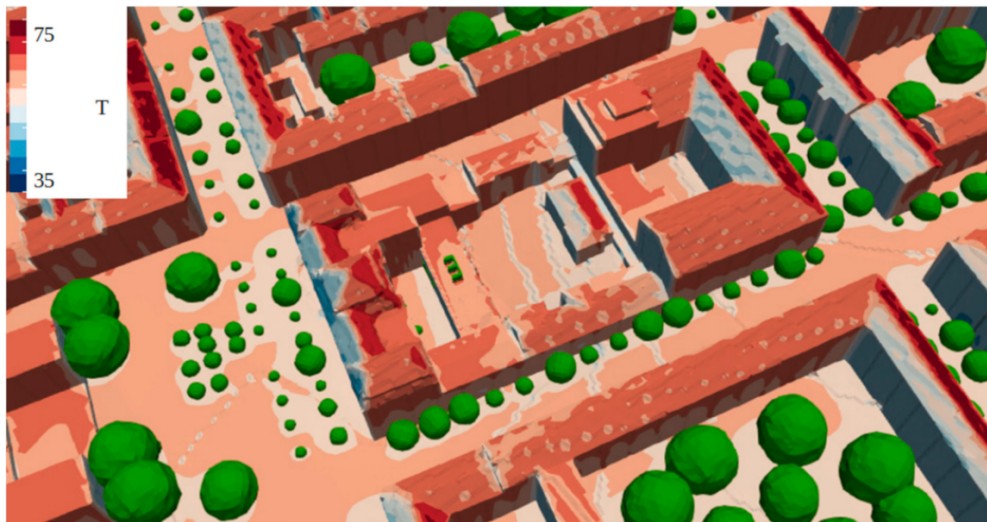

**Figure 27.** Surface temperatures (°C) around Schuhmeierplatz School on the model day at 3:00 p.m. CEST.

At Schuhmeierplatz School, the results of the simulation validation by means of the experimental data show a very good concordance (see Figure 29), with an average absolute error of −0.78 °C and a standard deviation of absolute errors of 0.74 °C at 3:00 p.m CEST. As there are no air temperature sensors directly exposed to solar radiation, the error due to the usage of non-ventilated radiation shields can be neglected. Further information on the validation results is provided in Appendix D.

The measurements and simulations were repeated in summer 2022 using ventilated radiation shields to be able to quantify the measurement error when using non-ventilated radiation shields. However, the results of the measurements in 2022 are not included in this study because the analysis of the data has not yet been completed. From the simulations performed so far, it is already apparent that a greening measure alone has very minor effects on the microclimate. To significantly reduce the risk of overheating in an inner courtyard or roof terrace in summer, greenery systems must be implemented on a much larger scale.

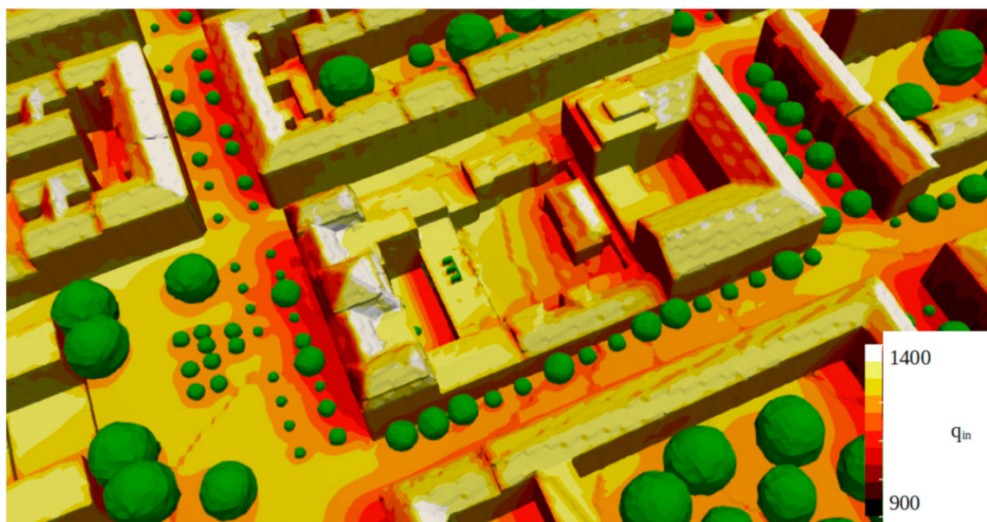

**Figure 28.** Incident radiation intensity (W/m$^2$) around Schuhmeierplatz School on the model day at 3:00 p.m. CEST.

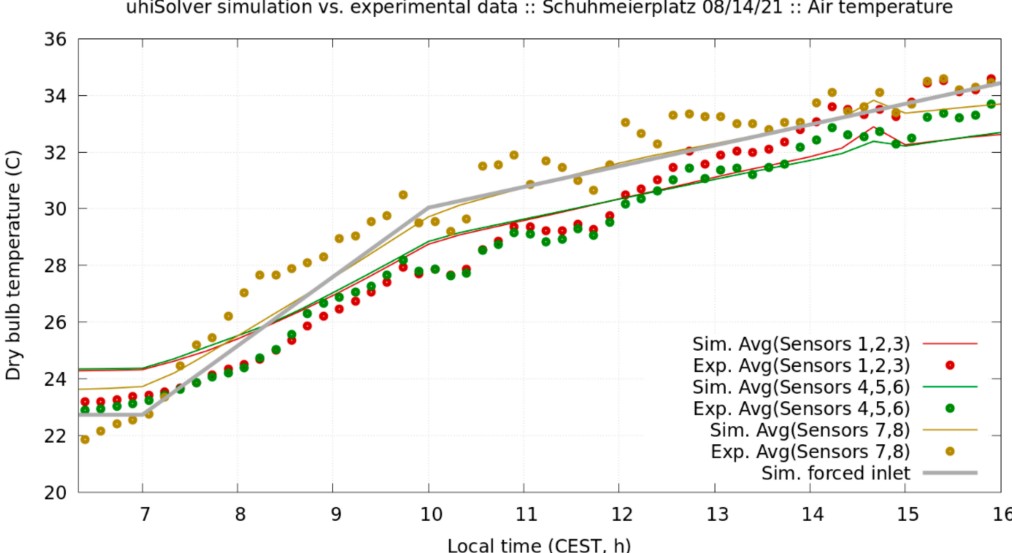

**Figure 29.** Validation of simulated air temperature with measured data at Schuhmeierplatz School on the model day from 06:20 a.m. to 4:00 p.m. CEST.

## 4. Discussion

In the present study, microclimate simulations are carried out with the software uhiSolver at two Viennese schools to investigate the influence of existing small-scale greenery systems in the inner courtyard and on the roof terrace, respectively, on the local microclimate. The simulation was validated with measurement data collected on-site. However, the measurement shows only minimal local effects of the greenery systems, even directly in front of the living walls and inside the green pergola. These insignificant effects are also seen in the closer free space in the simulation. These results are reproduced with tight intervals by uhiSolver. Hence, it is clearly shown that there is no significant local (0.5 m to 3 m distance from the wall) effect of small-scale living walls on thermodynamic and perceived temperatures on the outside, apart from building shading and lowered incident radiation behind the living wall (see Table 3). The green pergola is a special case, because of the larger shaded area compared to the living walls and proportionally bigger impact on outside apparent temperature. The biggest effect on apparent temperature is caused by the shading of the pergola, and not by air cooling due to evapotranspiration.

**Table 3.** Microclimatic effects of small-scale greenery on a hot summer day in Vienna.

|  | Air Temp. | Abs. Humidity | Shade | AT [1] |
|---|---|---|---|---|
| Living wall | No | Yes | Yes | No |
| Green pergola | No | Yes | Yes | Yes |

[1] Apparent temperature by Steadman [33].

The greening of facades is currently a dominant topic in Vienna and many other metropolitan areas around the world. The city of Vienna has set the goal of significantly increasing its number of green facades in the coming years [34] and has, therefore, launched various options for the funding of vertical greenery systems. Nevertheless, implementation is mostly limited to individual small-scale greening projects, as large-scale implementation is still associated with considerable costs. The care and maintenance of vertical greenery systems is often neglected. Furthermore, failures of the irrigation systems occur frequently, which is why many systems, especially living walls, are in a poor condition. The results of this study, which showed a maximum air temperature reduction of 0.3 °C at a 0.1 m distance from the north and north-west-facing living walls and within the green pergola, are supported by the findings of Daemei et al., who detected a peak temperate reduction of 0.36 °C at 12 p.m. in front of a north-facing direct green façade [22]. The same value of air temperature reduction was measured by Galagoda et al., at a 1 m distance from south to east-oriented living walls [27], indicating a wider range of the cooling effect at vertical greenery systems that are more exposed to solar radiation.

Therefore, it seems obvious that a single vertical greenery system covering an area of just a few square meters is not enough to reduce the ambient air temperature and increase thermal comfort in summer. The greening of facades will have to be applied on a much larger scale, with estimated air temperature reductions of 0.3 °C when applying green facades with a coverage fraction of 50% within an east-west-oriented mid-rise low-density street canyon [24], and about 1 °C at a high-rise high-density scenario with 100% of the facades of a building block being covered with living walls [26]. Adding vegetation as avenue trees could lead to higher reductions in air temperature of up to 2 °C, especially in the shadow of the trees [23,24]. To substantially contribute to a reduction in the UHI effect in big cities, a combination of greenery systems with other microclimatic measures is needed, such as, for example, water sprays and fountains, enabling a decrease in the local air temperature by a maximum of 7 °C [23]. Furthermore, the influence of the increased humidity on the apparent (perceived) temperature must also be taken into account. At low wind speeds, the increased humidity of the vertical greenery system and/or water sprays will not be transported away and may lead to an increase in the perceived temperature, despite the air temperature being lowered. To ensure that an implemented greenery system has the desired effect on the microclimate, a combined consideration of the different effects of the greenery on the air temperature, the air humidity, and the surface temperature, as well as the incident surface radiation intensity must, therefore, be carried out.

Contributions to the fields of UHI research and policies of this study are as follows:

*Scientific findings*:

- It is clearly shown by both experimental and simulation data that small-scale greening measures—especially on north-facing or mostly shaded facades—cannot fulfill the expectations of any practical outside air cooling;
- The anticipated positive influence on microclimate is not achieved under a certain size (area, leaf area index, crown diameter, etc.) of the greening, which evidently has to be much larger than in the studied greenings;
- Providing convincing data on the ineffectiveness of some greening systems documents the root causes of failed implementations and incentivizes future research into apt measures in urban climate adaptation.

*Recommendations for policy makers*:

- Avoiding funding and construction of greening systems that are scientifically proven to be ineffective measures against overheating urban areas;
- In order to improve outside thermal comfort in summer, city authorities should focus on more differentiated, effective greening and shading measures instead of piecemeal living walls;
- Only perfectly maintained large-scale greening will provide some felt air cooling effect and, more importantly, reduce apparent temperatures and provide sufficient shade for pedestrians and buildings.

## 5. Conclusions

At two Viennese school buildings, north to northwest-facing living walls and a green pergola were examined. Transient dynamic microclimate simulations with the uhiSolver software were carried out that show a very small impact of the greening measures on the local microclimate, rapidly decreasing with increasing distance from the greenery. Compared to a bare wall, the maximum decrease in ambient air temperature at 0.1 m from the façade greening is 0.3 °C on a hot summer day at 3:00 p.m. local time (CEST). The same reduction in air temperature is obtained within the green pergola, whereas the apparent (perceived) temperature is reduced by up to 4 °C, mostly due to the shading of the green pergola compared to the unshaded roof terrace. A comparison with on-site measurement data shows a good agreement.

City departments' actions to mitigate UHI effects should simultaneously incorporate differentiated strategies and include the large-scale application of vertical greenery systems, green roofs—where possible and sensible—and large crown avenue trees. However, further research is needed to assess the influence of different vertical greenery systems on the local microclimate and to examine to what extent they may be able to contribute to mitigating UHI formation when applied at a city-scale.

**Author Contributions:** F.T.: conceptualization, investigation, methodology, resources, writing—original draft preparation, project administration. A.H. and M.L.: software, data curation, validation, visualization. A.K.: writing—review and editing, supervision, funding acquisition. All authors have read and agreed to the published version of the manuscript.

**Funding:** This research was funded by Klima- und Energiefonds and Österreichische Forschungsförderungsgesellschaft FFG, grant number 884762.

**Institutional Review Board Statement:** Not applicable.

**Data Availability Statement:** The data that support the findings of this study are available from Rheologic GmbH but restrictions apply to the availability of these data, which were used under license for the current study, and so are not publicly available. Data are however available from the authors upon reasonable request and with permission of Rheologic GmbH.

**Acknowledgments:** This research was supported by the Austrian Research Promotion Agency (FFG) as well as by the Climate and Energy Fund of the Federal Ministry for Climate Action, Environment, Energy, Mobility, Innovation and Technology (BMK) within the framework of the Smart Cities Demo—Boosting Urban Innovation funding line. Open Access Funding by TU Wien.

**Conflicts of Interest:** The authors declare no conflict of interest. The funders had no role in the design of the study; in the collection, analyses, or interpretation of data; in the writing of the manuscript, or in the decision to publish the results.

# Appendix A

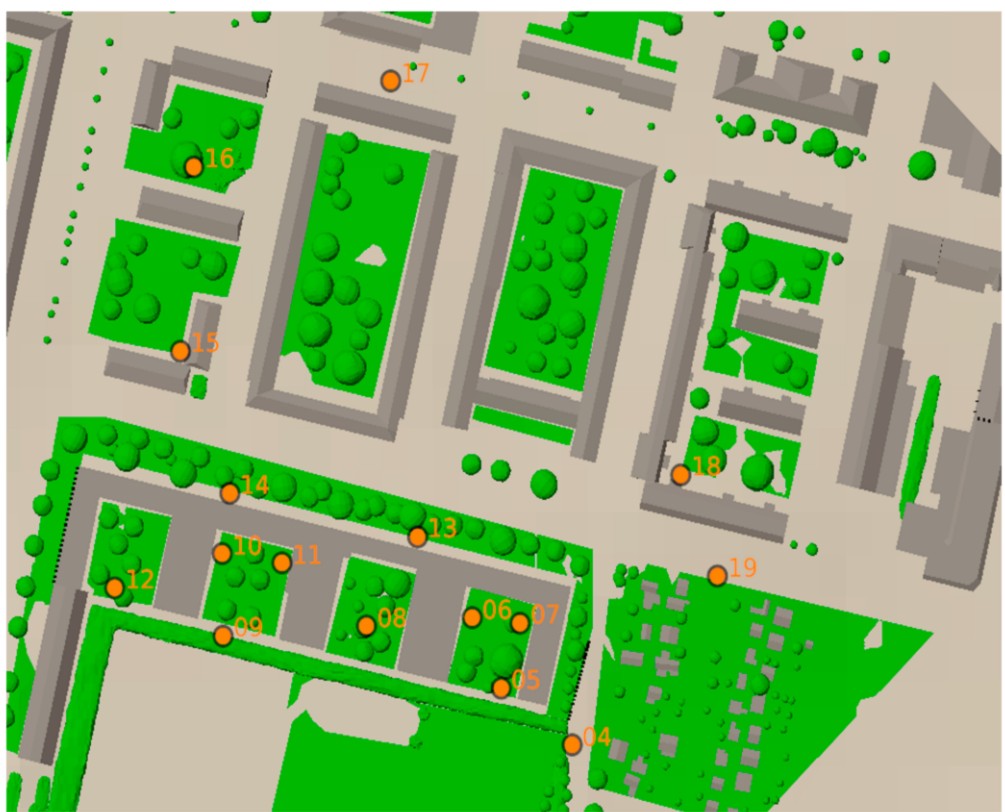

**Figure A1.** Sample positions at the 16.31954 E, 48.20599 N geo-coordinates for the validation case of Gablenzgasse.

**Table A1.** The uhiSolver simulation results for the validation case of Gablenzgasse, Vienna, on 29 June 2021 at 15:00 to 16:10 CEST.

| Pos. (1) | Time (CEST) | uhiSolver Simulation Results | | | | | |
|---|---|---|---|---|---|---|---|
| | | U (m/s) | T (Air) (K) | T (Air) (C) | RH (1) | AT (°C) | X (g/kg) |
| 04 | | 0.95 | 303.6 | 30.5 | 0.43 | 39.36 | 11.937 |
| 05 | 15:00 | 0.07 | 303.2 | 30.1 | 0.45 | 40.88 | 12.120 |
| 06 | | 0.13 | 303.0 | 29.8 | 0.46 | 40.22 | 12.301 |
| 07 | | 0.24 | 302.9 | 29.7 | 0.46 | 40.01 | 12.112 |
| 08 | 15:10 | 0.16 | 302.9 | 29.7 | 0.46 | 40.19 | 12.206 |
| 09 | 15:15 | 0.31 | 303.2 | 30.0 | 0.45 | 40.12 | 12.053 |
| 10 | | 0.17 | 302.9 | 29.7 | 0.46 | 39.90 | 12.087 |
| 11 | | 0.33 | 303.2 | 30.1 | 0.44 | 40.26 | 11.933 |
| 12 | 15:30 | 0.31 | 303.0 | 29.8 | 0.45 | 39.81 | 12.019 |
| 13 | | 0.24 | 302.3 | 29.2 | 0.47 | 39.05 | 12.082 |
| 14 | 15:45 | 0.03 | 302.4 | 29.2 | 0.47 | 39.53 | 12.124 |
| 15 | | 0.22 | 302.8 | 29.6 | 0.45 | 39.37 | 11.837 |
| 16 | | 0.15 | 302.5 | 29.4 | 0.47 | 39.61 | 12.078 |
| 17 | 16:00 | 0.07 | 302.4 | 29.3 | 0.46 | 39.60 | 11.891 |
| 18 | 16:10 | 0.11 | 301.9 | 28.7 | 0.49 | 38.83 | 12.167 |
| 19 | | 0.56 | 303.2 | 30.0 | 0.44 | 39.89 | 11.933 |

**Table A2.** Experimental data for the validation case of Gablenzgasse, Vienna, on 29 June 2021 at 15:00 to 16:10 CEST.

| Pos. (1) | Time (CEST) | Experimental Data | | | | | |
|---|---|---|---|---|---|---|---|
| | | U min (m/s) | U max (m/s) | U avg (m/s) | T (air) (C) | RH (%) | X (g/kg) |
| 04 | | 0.20 | 0.20 | 0.20 | 31.2 | 41.7 | 12.000 |
| 05 | 15:00 | 0.20 | 0.30 | 0.25 | 29.8 | 45.1 | 11.953 |
| 06 | | 0.10 | 0.10 | 0.10 | 29.5 | 45.1 | 11.737 |
| 07 | | 0.30 | 0.30 | 0.30 | 32.3 | 41.0 | 12.577 |
| 08 | 15:10 | 0.50 | 0.50 | 0.50 | 29.8 | 45.3 | 12.047 |
| 09 | 15:15 | 0.16 | 0.20 | 0.18 | 29.8 | 44.7 | 11.848 |
| 10 | | 0.10 | 0.20 | 0.15 | 30.1 | 45.7 | 12.362 |
| 11 | | 0.15 | 0.15 | 0.15 | 32.5 | 39.2 | 12.152 |
| 12 | 15:30 | 0.30 | 0.45 | 0.38 | 29.1 | 45.7 | 11.625 |
| 13 | | 0.60 | 1.20 | 0.90 | 29.5 | 44.6 | 11.648 |
| 14 | 15:45 | 0.70 | 0.95 | 0.83 | 30.1 | 43.5 | 11.753 |
| 15 | | 0.20 | 0.30 | 0.25 | 30.0 | 44.6 | 11.957 |
| 16 | | 1.50 | 1.80 | 1.65 | 29.3 | 46.8 | 12.072 |
| 17 | 16:00 | 0.50 | 0.50 | 0.50 | 30.4 | 41.8 | 11.496 |
| 18 | 16:10 | 0.20 | 0.45 | 0.33 | 29.2 | 46.2 | 11.871 |
| 19 | | 0.20 | 0.50 | 0.35 | 30.7 | 41.0 | 11.488 |

**Table A3.** Absolute error between the uhiSolver simulation results and experimental data for the validation case of Gablenzgasse, Vienna, on 29 June 2021 at 15:00 to 16:10 CEST.

| Pos. (1) | Time (CEST) | uhiSolver Absolute Error | | |
|---|---|---|---|---|
| | | U (m/s) | T (K) | X (g/kg) |
| 04 | | 0.75 | −0.7 | −0.063 |
| 05 | 15:00 | −0.18 | 0.3 | 0.167 |
| 06 | | 0.03 | 0.3 | 0.564 |
| 07 | | −0.06 | −2.6 | −0.466 |
| 08 | 15:10 | −0.34 | −0.1 | 0.160 |
| 09 | 15:15 | 0.13 | 0.3 | 0.205 |
| 10 | | 0.02 | −0.4 | −0.275 |
| 11 | | 0.18 | −2.4 | −0.218 |
| 12 | 15:30 | −0.06 | 0.8 | 0.394 |
| 13 | | −0.66 | −0.3 | 0.435 |
| 14 | 15:45 | −0.79 | −0.9 | 0.371 |
| 15 | | −0.03 | −0.3 | −0.120 |
| 16 | | −1.50 | 0.1 | 0.007 |
| 17 | 16:00 | −0.43 | −1.2 | 0.395 |
| 18 | 16:10 | −0.22 | −0.5 | 0.296 |
| 19 | | 0.21 | −0.7 | 0.445 |

**Table A4.** Averaged absolute errors and standard deviation of absolute errors for the validation case of Gablenzgasse, Vienna, on 29 June 2021 at 15:00 to 16:10 CEST.

| Averaged Absolute Errors | | |
|---|---|---|
| U (m/s) | T (K) | X (g/kg) |
| −0.18 | −0.53 | 0.143 |
| **Std. Dev. of Absolute Errors** | | |
| U (m/s) | T (K) | X (g/kg) |
| 0.50 | 0.93 | 0.302 |

## Appendix B

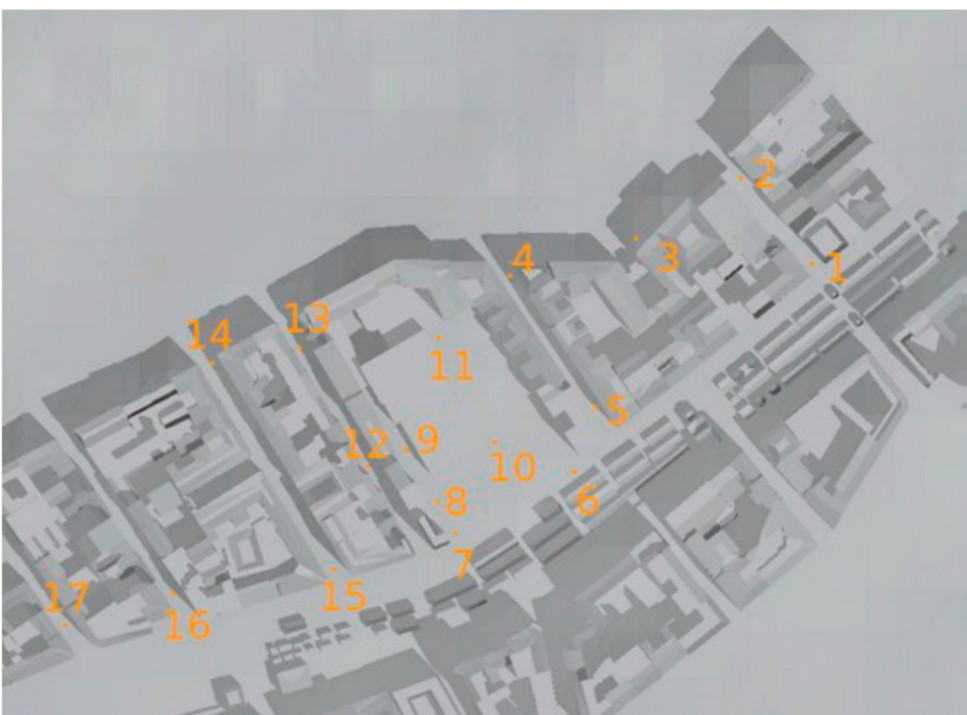

**Figure A2.** Sample positions at the 16.36237 E, 48.19798 N geo-coordinates for the validation case of Naschmarkt.

**Table A5.** The uhiSolver simulation results for the validation case of Naschmarkt, Vienna, on 21 June 2022 at 14:00 to 15:55 CEST.

| Pos. | Time | uhiSolver Simulation Results | | | | | |
|------|------|---------|-----------|-----------|--------|---------|---------|
| (1) | (CEST) | U (m/s) | T (Air) (K) | T (Air) (C) | RH (1) | AT (°C) | X (g/kg) |
| 01 | 14:00 | 1.4 | 304.9 | 31.7 | 0.45 | 40.2 | 13.45 |
| 02 | | 0.9 | 305.4 | 32.2 | 0.43 | 41.6 | 13.45 |
| 03 | | 0.6 | 305.8 | 32.7 | 0.42 | 44.1 | 13.45 |
| 04 | | 0.5 | 305.8 | 32.7 | 0.42 | 42.8 | 13.47 |
| 05 | | 1.2 | 305.7 | 32.5 | 0.43 | 41.2 | 13.48 |
| 06 | | 0.2 | 306.1 | 32.9 | 0.43 | 45.2 | 13.76 |
| 07 | | 0.0 | 304.9 | 31.7 | 0.46 | 43.0 | 13.78 |
| 08 | | 0.1 | 304.3 | 31.2 | 0.48 | 42.2 | 13.88 |
| 09 | 15:00 | 0.1 | 303.2 | 30.0 | 0.52 | 41.4 | 14.37 |
| 10 | | 0.2 | 305.6 | 32.5 | 0.45 | 44.6 | 14.09 |
| 11 | 15:10 | 0.1 | 304.6 | 31.4 | 0.48 | 42.7 | 14.14 |
| 12 | | 0.2 | 304.1 | 30.9 | 0.47 | 41.7 | 13.60 |
| 13 | | 0.9 | 305.2 | 32.1 | 0.44 | 41.5 | 13.47 |
| 14 | | 1.4 | 305.7 | 32.5 | 0.43 | 40.8 | 13.45 |
| 15 | 15:40 | 0.5 | 304.8 | 31.6 | 0.45 | 42.0 | 13.47 |
| 16 | | 2.0 | 305.4 | 32.3 | 0.43 | 39.7 | 13.44 |
| 17 | 15:55 | 2.3 | 305.9 | 32.8 | 0.42 | 39.6 | 13.44 |

**Table A6.** Experimental data for the validation case of Naschmarkt, Vienna, on 21 June 2022 at 14:00 to 15:55 CEST.

| Pos. | Time | Experimental Data | | | | | |
|---|---|---|---|---|---|---|---|
| (1) | (CEST) | U min (m/s) | U max (m/s) | U avg (m/s) | T (Air) (C) | RH (%) | X (g/kg) |
| 01 | 14:00 | 1.2 | 2.6 | 1.9 | 31.0 | 33.3 | 10.77 |
| 02 | | 1.5 | 1.5 | 1.5 | 31.0 | 32.6 | 10.13 |
| 03 | | 0.9 | 0.9 | 0.9 | 30.1 | 30.6 | 8.46 |
| 04 | | 0.3 | 0.7 | 0.5 | 31.1 | 32.9 | 10.40 |
| 05 | | 0.8 | 0.8 | 0.8 | 31.7 | 32.3 | 9.86 |
| 06 | | 1.1 | 1.1 | 1.1 | 31.7 | 29.5 | 7.65 |
| 07 | | 0.2 | 0.2 | 0.2 | 30.9 | 30.3 | 8.23 |
| 08 | | 0.5 | 0.5 | 0.5 | 30.7 | 33.0 | 10.49 |
| 09 | 15:00 | 0.5 | 1.0 | 0.8 | 31.7 | 33.5 | 10.96 |
| 10 | | 0.3 | 0.7 | 0.5 | 31.6 | 32.3 | 9.86 |
| 11 | 15:10 | 0.3 | 0.3 | 0.3 | 30.1 | 35.8 | 13.36 |
| 12 | | 0.7 | 0.7 | 0.7 | 32.1 | 31.6 | 9.26 |
| 13 | | 0.7 | 0.7 | 0.7 | 32.2 | 31.7 | 9.35 |
| 14 | | 1.1 | 1.1 | 1.1 | 34.0 | 29.6 | 7.72 |
| 15 | 15:40 | 0.7 | 2.1 | 1.4 | 33.4 | 29.5 | 7.65 |
| 16 | | 0.4 | 1.4 | 0.9 | 33.1 | 29.3 | 7.51 |
| 17 | 15:55 | 0.7 | 2.5 | 1.6 | 32.5 | 31.3 | 9.02 |

**Table A7.** Absolute error between the uhiSolver simulation results and the experimental data for the validation case of Naschmarkt, Vienna, on 21 June 2022 at 14:00 to 15:55 CEST.

| Pos. | Time | uhiSolver Absolute Error | | |
|---|---|---|---|---|
| (1) | (CEST) | U (m/s) | T (K) | X (g/kg) |
| 01 | 14:00 | −0.5 | 0.7 | 2.68 |
| 02 | | −0.6 | 1.2 | 3.32 |
| 03 | | −0.3 | 2.6 | 4.99 |
| 04 | | 0.0 | 1.6 | 3.07 |
| 05 | | 0.4 | 0.8 | 3.62 |
| 06 | | −0.9 | 1.2 | 6.11 |
| 07 | | −0.2 | 0.8 | 5.55 |
| 08 | | −0.4 | 0.5 | 3.39 |
| 09 | 15:00 | −0.7 | −1.7 | 3.41 |
| 10 | | −0.3 | 0.9 | 4.23 |
| 11 | 15:10 | −0.2 | 1.3 | 0.78 |
| 12 | | −0.5 | −1.2 | 4.34 |
| 13 | | 0.2 | −0.1 | 4.12 |
| 14 | | 0.3 | −1.5 | 5.74 |
| 15 | 15:40 | −0.9 | −1.8 | 5.83 |
| 16 | | 1.1 | −0.8 | 5.93 |
| 17 | 15:55 | 0.7 | 0.3 | 4.43 |

**Table A8.** Averaged absolute errors and standard deviation of absolute errors for the validation case of Naschmarkt, Vienna, on 21 June 2022 at 14:00 to 15:55 CEST.

| Averaged Absolute Errors | | |
|---|---|---|
| U (m/s) | T (K) | X (g/kg) |
| −0.19 | 0.09 | 4.19 |
| **Std. Dev. of Absolute Errors** | | |
| U (m/s) | T (K) | X (g/kg) |
| 0.63 | 1.13 | 1.89 |

**Appendix C**

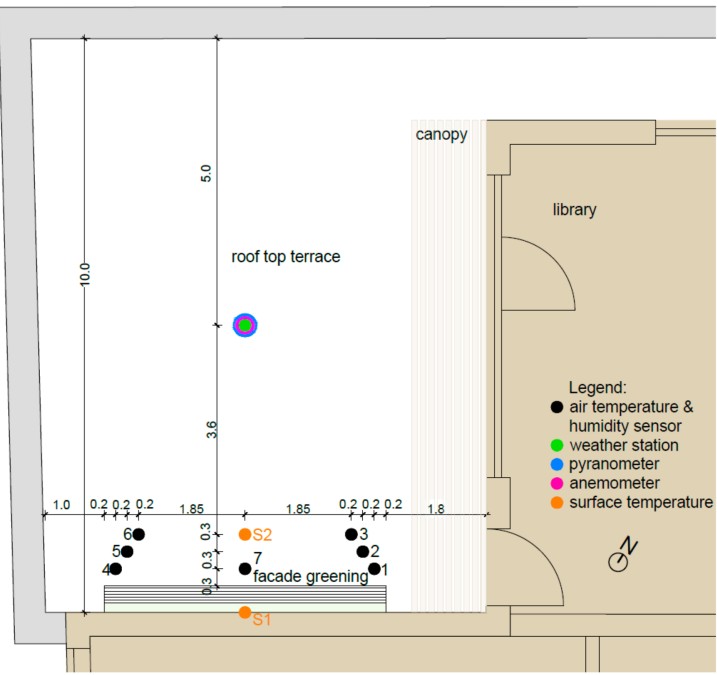

**Figure A3.** Sample positions at the 16.33293 E, 48.18555 N geo-coordinates for the Diefenbachgasse School with the air temperature and humidity sensors 1 to 7 and the surface temperature sensors S1 and S2.

**Table A9.** The uhiSolver simulation results for Diefenbachgasse School on the model day at 15:00 CEST.

| Pos. | uhiSolver Simulation Results | | | | |
|------|------|------|------|------|------|
| (1) | U (m/s) | Wind-Dir. (-) | T (Air) (°C) | RH (%) | X (g/kg) |
| 1 | 0.4 | WSW | 31.2 | 47 | 0.0136 |
| 2 | 0.4 | WSW | 31.2 | 47 | 0.0136 |
| 3 | 0.4 | WSW | 31.2 | 47 | 0.0136 |
| 4 | 0.6 | WSW | 31.5 | 45 | 0.0135 |
| 5 | 0.6 | WSW | 31.7 | 45 | 0.0135 |
| 6 | 0.6 | WSW | 31.7 | 45 | 0.0135 |
| 7 | 0.6 | WSW | 31.7 | 44 | 0.0135 |
| W.S. | 0.6 | WSW | 31.6 | 45 | 0.0135 |

**Table A10.** Experimental data for Diefenbachgasse School on the model day at 15:00 CEST.

| Pos. | Experimental Data | | | | | | |
|------|------|------|------|------|------|------|------|
| (1) | U min (m/s) | U max (m/s) | U avg (m/s) | Wind-Dir. (-) | T (Air) (°C) | RH (%) | X (kg/kg) |
| 1 | - | - | - | - | 34.2 | 35 | 0.0115 |
| 2 | - | - | - | - | 35.0 | 33 | 0.0120 |
| 3 | - | - | - | - | 35.0 | 32 | 0.0120 |
| 4 | - | - | - | - | 34.4 | 33 | 0.0110 |
| 5 | - | - | - | - | 32.0 | 31 | 0.0090 |
| 6 | - | - | - | - | 34.7 | 29 | 0.0090 |
| 7 | - | - | - | - | 33.9 | 32 | 0.0105 |
| W.S. | 0.4 | 3.2 | 1.6 | SW | 34.8 | 37 | 0.0130 |

**Table A11.** Absolute error between the uhiSolver simulation results and the experimental data for Diefenbachgasse School on the model day at 15:00 CEST.

| Pos. | uhiSolver Absolute Error | | |
|---|---|---|---|
| (1) | U (m/s) | T (K) | X (kg/kg) |
| 1 | - | −3.0 | 0.0021 |
| 2 | - | −3.8 | 0.0016 |
| 3 | - | −3.8 | 0.0016 |
| 4 | - | −2.9 | 0.0025 |
| 5 | - | −0.3 | 0.0045 |
| 6 | - | −3.0 | 0.0045 |
| 7 | - | −2.2 | 0.0030 |
| W.S. | −1.0 | −3.2 | 0.0005 |

**Table A12.** Averaged absolute errors and standard deviation of absolute errors for Diefenbachgasse School on the model day at 15:00 CEST.

| Averaged Absolute Errors | | |
|---|---|---|
| U (m/s) | T (K) | X (kg/kg) |
| −1.0 | −2.78 | 0.0025 |
| **Std. Dev. of Absolute Errors** | | |
| U (m/s) | T (K) | X (kg/kg) |
| - | 1.12 | 0.0014 |

**Appendix D**

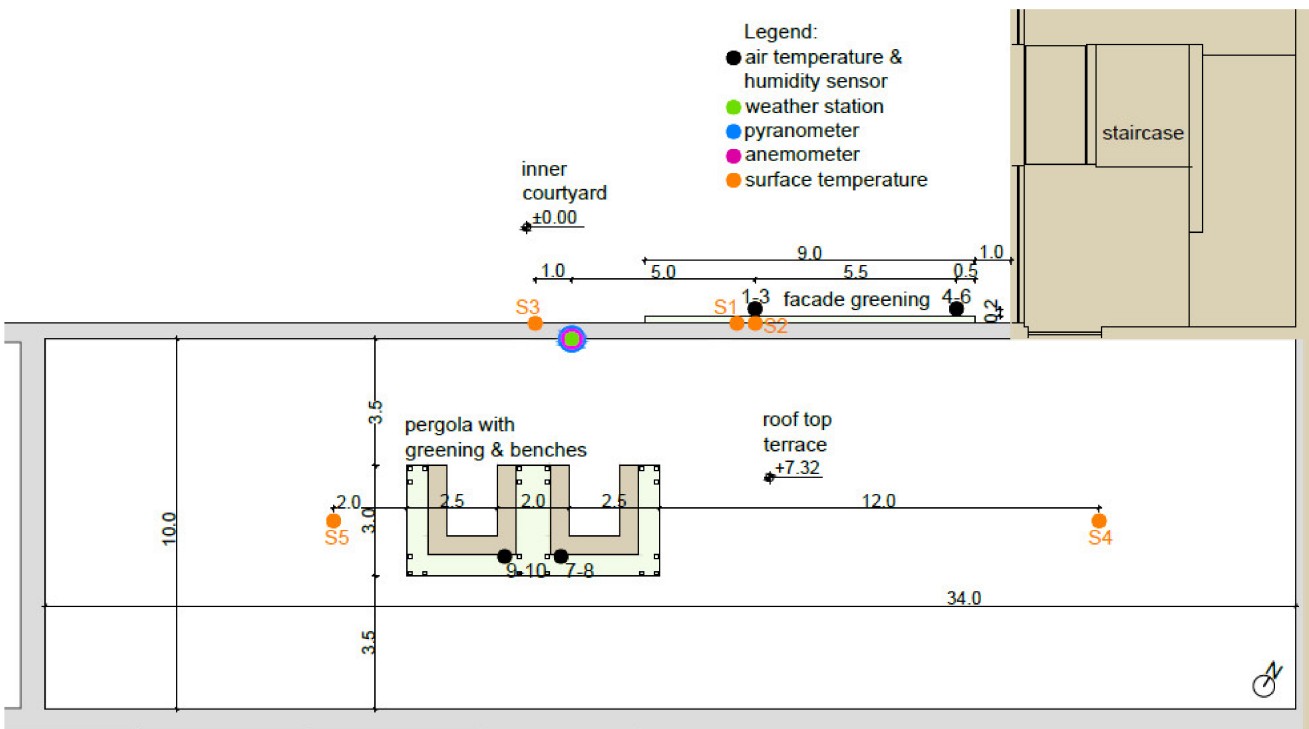

**Figure A4.** Sample positions at the 16.31892 E, 48.20989 N geo-coordinates for the Schuhmeierplatz School with the air temperature and humidity sensors 1 to 10 and the surface temperature sensors S1 to S5.

**Table A13.** The uhiSolver simulation results for Schuhmeierplatz school on the model day at 15:00 CEST.

| Pos. | uhiSolver Simulation Results | | | | |
|------|--------|--------------|-------------|--------|------------|
| (1) | U (m/s) | Wind-Dir. (-) | T (Air) (°C) | RH (%) | X (kg/kg) |
| 1 | 0.3 | E | 32.3 | 44 | 0.0136 |
| 2 | 0.3 | E | 32.3 | 44 | 0.0136 |
| 3 | 0.3 | E | 32.3 | 44 | 0.0136 |
| 4 | 0.3 | E | 32.1 | 44 | 0.0135 |
| 5 | 0.3 | E | 32.2 | 44 | 0.0135 |
| 6 | 0.3 | E | 32.3 | 43 | 0.0135 |
| 7 | 0.2 | NNE | 33.4 | 41 | 0.0136 |
| 8 | 0.2 | NNE | 33.4 | 41 | 0.0136 |
| W.S. | 0.3 | ESE | 32.6 | 43 | 0.0135 |

**Table A14.** Experimental data for Schuhmeierplatz school on the model day at 15:00 CEST.

| Pos. | Experimental Data | | | | | | |
|------|-------------|-------------|-------------|--------------|-------------|--------|-----------|
| (1) | U min (m/s) | U max (m/s) | U avg (m/s) | Wind-Dir. (-) | T (Air) (°C) | RH (%) | X (kg/kg) |
| 1 | - | - | - | - | 33.0 | 43 | 0.0140 |
| 2 | - | - | - | - | 34.5 | 41 | 0.0145 |
| 3 | - | - | - | - | 33.8 | 43 | 0.0145 |
| 4 | - | - | - | - | 32.5 | 43 | 0.0135 |
| 5 | - | - | - | - | 32.7 | 42 | 0.0135 |
| 6 | - | - | - | - | 32.3 | 45 | 0.0130 |
| 7 | - | - | - | - | 34.2 | 40 | 0.0135 |
| 8 | - | - | - | - | 33.2 | 43 | 0.0135 |
| W.S. | 3.2 | 4.8 | 4.0 | ESE | 33.7 | 40 | 0.0133 |

**Table A15.** Absolute error between the uhiSolver simulation results and the experimental data for Schuhmeierplatz School on the model day at 15:00 CEST.

| Pos. | uhiSolver Absolute Error | | |
|------|---------|--------|-----------|
| (1) | U (m/s) | T (K) | X (kg/kg) |
| 1 | - | −0.7 | −0.0004 |
| 2 | - | −2.3 | −0.0009 |
| 3 | - | −1.5 | −0.0009 |
| 4 | - | −0.4 | 0.0000 |
| 5 | - | −0.5 | 0.0000 |
| 6 | - | 0.1 | 0.0005 |
| 7 | - | −0.8 | 0.0001 |
| 8 | - | 0.2 | 0.0001 |
| W.S. | −3.7 | −1.1 | 0.0002 |

**Table A16.** Averaged absolute errors and standard deviation of absolute errors for Schuhmeierplatz School on the model day at 15:00 CEST.

| Averaged Absolute Errors | | |
|---------|--------|-----------|
| U (m/s) | T (K) | X (kg/kg) |
| −3.7 | −0.78 | 0.0001 |

| Std. Dev. of Absolute Errors | | |
|---------|--------|-----------|
| U (m/s) | T (K) | X (kg/kg) |
| - | 0.74 | 0.0005 |

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
