# Peer review of "The Impact of Small-Scale Greening on the Local Microclimate—A Case Study at Two School Buildings in Vienna"

_sustainability, doi:10.3390/su142013089_

Round 1

Reviewer 1 Report

The authors should elaborate the theoretical background in the part 1.1. since in its current form it provides only a list of general information on the topic. Additionally, segments 1.2. and 1.3. should include some international examples with a similar focus.

The discussion should be restructured - the elements of literature review, if relevant in the theoretical sense, should be included into the part 1.1., while the selected cases, used for the comparison, should be analysed (and related to the selected Vienna schools) in accordance to the pre-defined set of criteria.

The conclusion could include some recommendations derived from successful international/national examples.

Author Response

Thank you for reading and commenting our publication draft.

For point-by-point response to your comments please see the attachment.

Reviewer 2 Report

The impact of building greening on the local microclimate – a case study at two school buildings in Vienna

Review

 Summary

This article presents the materials and setup of two small north-facing green walls and some vines on a rooftop on two university campuses and discusses their potential to cool urban environments. The green walls, although described in the methods, are not reported directly. It is not clear how they are used in the study since there is conflicting information. Lines 123-124 state that the data was not used while lines 501-503 state they were? Results are presented for macroscale effects of green walls, from modeling, but without a detailed description of the software, theory of how or why it is a valid model or detail on the modeling setup. Results and discussion refer to the potential wind and thermal comfort potential of green walls discussed in the methods. Overall, the article is difficult to follow and concludes that the walls in the setup are not effective for cooling multi-building sites. The research topic is interesting and potentially useful; however, the research setup and confusing organization of the article and its conflicting statements and lack of information place the article in need of a re-write. Why is it interesting or important to the research community that small north-facing green walls don’t contribute to overall reducing urban heat islands? From the literature review and discussion, it is not evident to expect small north-facing green walls to have an effect. Why is reporting this research important?

Detailed Comments

1.2. Previous research projects on the influence of building greening

This section should reach out to literature beyond the school, and to existing studies elsewhere to help set up the research goals and question. Why is the literature review included in section 4? Some content is necessary to set up the research question. Line 109-113 explains what was done, but not why (in the context of existing literature). Existing research on the microclimate effects of green walls already establishes that when used in a largescale application on west-facing facades, green walls can reduce air temperatures by up to one-third. This paper below establishes a valid method and set up to collect data at the meta scale. See International Journal of Low-Carbon Technologies, Volume 10, Issue 1, March 2015, Pages 34–44, Experimental study of the urban microclimate mitigation potential of green roofs and green walls in street canyons, Rabah Djedjig, Emmanuel Bozonnet, Rafik Belarbi

Why and how are north-facing green walls expected to benefit urban heat islands? The introduction does not address this issue.

Methods

The experimental setup in the Vienna research describes methods for a microclimate study of two small north-facing living walls and one climbing vine (Fig 7 shows only a few mature vines while Fig 16 shows lush coverage). However, where/how are local microclimate results presented in this paper? There is a mismatch between the title of the paper, the description provided in the methods and materials, and the content presented in the results and discussion. Perhaps a table or graph is necessary to demonstrate the observed temperatures and humidity of the green walls and vines and how it was used in the model.

Lines 123-126 need much expansion and explanation. If the green walls described in sections 2.1 and 2.2 are not used to observe and validate the model, then why are they included? Why do lines 501-503 say this data was used?

Line 149. Much detail is lacking. What is the make and model of sensors? How was data from sensors 1-7 (for the green wall) used? How were the data incorporated into the model? Make and model of the weather station. How far are these devices located from the wall?

There are no methods provided to discuss how someone could replicate the setup and validate the model?

The green wall at the Diefenbachgasse school is only about 60-65 percent vegetated (Fig. 2). The black background of green wall material is partially vegetated. The green wall surface is not mature or representative of a healthy green wall. However, the wall faces northwest? North of the equator, the most direct benefit of a green wall would be south to west facing orientation. None of the experimental setups in the Vienna study face south or west. The study is largely observing the effect of shaded living walls. It is not clear why only north-facing and shaded green walls were used to validate the model. 

Irrigation rates and frequency are missing for both green walls.

Based on the small size of the living walls (compared to the surface area of all walls), there is no rationale why one would expect that a small-scale partially vegetated north-facing green wall could potentially improve or reduce the air temperature of the multi-building site. Djedjig et al, already demonstrate that large applications of green walls are necessary to expect some effect when facing west.

Impact of urban heat island mitigation measures on microclimate and pedestrian comfort in a dense urban district of Lebanon, Jeff Faheda Elias Kinabb Stephane Ginesteta LucAdolphec.  This paper is cited in the manuscript. It is excellent. The methods and research outcomes provide a good example of how to set up materials and methods that are linked to the relevant results and title.

Results

Lines 219-220 "The goal of this wind simulation was to determine the large-scale wind direction that matches the wind conditions on the rooftop terrace according to the measured data from the weather station." There needs to be a discussion in the introduction to set up and make a statement about the research goal. Literature review and methods should support the goal of the research.

Overall, the methods are not suitable for the scope of the research presented. The results are not reproducible from the methods and setup. The green walls described in the methods (but not used) are too small to expect any effect and their position facing north poses little to no potential benefit. It is not communicated why the experimental setup is suitable for the proposed outcome of the study. It is already intuitive that a small north-facing green wall would not be expected to modify the microclimate of an urban canyon.

Lines 249-251 proved the exact locations of sensors. Figure 1 in methods lacks the exact locations of sensors on the y-axis.

Recommendations

Re-write the article to make the research goals clear, validate methods, and report observations. Address issues reported in the reviews.

Check out these papers

Building and Environment, Volume 181, 15 August 2020, 106923, Assessment of the effect of living wall systems on the improvement of the urban heat island phenomenon, Elham Shafiee, Mohsen Faizi. Seyed-Abbas Yazdanfar. Mohammad-Ali Khanmohammadi

International Journal of Low-Carbon Technologies, Volume 10, Issue 1, March 2015, Pages 34–44, Experimental study of the urban microclimate mitigation potential of green roofs and green walls in street canyons, Rabah Djedjig, Emmanuel Bozonnet, Rafik Belarbi

Renewable and Sustainable Energy Reviews, Volume 159, May 2022, 112100   Effect of green wall installation on urban heat island and building energy use: A climate-informed systematic literature review, Susca F. ZanghirellaL. Colasuonno V. Del Fatto

Author Response

Thank you for reading and detailled commenting our publication draft.

For point-by-point response to your comments please see the attachment.

Reviewer 3 Report

The paper reports on simulation results of the effect of different building greening measures on the local microclimate. The paper is well structured, however I would like to address several minor comment to improve the quality of the paper:

     Building vertical greenery systems are commonly categorized in green façades and living walls, mainly in relation to the location of growing media. Please see the classification reported in Safikhani et al., 2014 (Safikhani, T., Abdullah, A.M., Ossen, D.R., & Baharvand, M., A review of energy characteristic of vertical greenery systems. Renewable and Sustainable Energy Reviews, 40, pp. 450–462, 2014. https://doi.org/10.1016/j.rser.2014.07.166). It would add value to the text to use more appropriate terminology right from the abstract to the conclusion. This will facilitate understanding of the text and facilitate comparison of results by the scientific community.

     Building greenery systems thermal effects vary depending on several factors including building characteristics, local climate, exposition, plants characteristics, type of greenery system. Section 2 should provide more details on the examined greenery systems (plant species, substrate and vegetation layers thickness, leaf area index, presence of an air gap between the building walls and the vegetation vertical layer and related thickness), on the local climate in the examined period (a table could be useful for showing average, maximum and minimum values of air temperature, wind velocity and direction, air relative humidity).

     It would be interesting to understand whether the wind conditions on the day under consideration in each case study are representative of the examined period. Please specify. Otherwise, it would be appropriate to study the results of different simulations performed considering different environmental conditions.

     It would be appropriate to provide the results of the simulation validation against the experimental data.

     Whenever possible, avoid the use of the generic term "façade greening" in the discussion section and specify whether it is the case of a green façade or a living wall.

Author Response

(The authors gave the same response as above.)

Round 2

Reviewer 1 Report

The revised version of the article is significantly improved, in accordance with the suggestions. 

Author Response

Thank you for your comments!

Extensive English spell check has been done to improve the quality of the paper.

Reviewer 2 Report

While many of the comments regarding clarity were addressed, the value of the findings is not greatly improved regarding urban heat islands. There is a need for research on green facades regarding understanding the value and density of their use on south and west-facing slopes in the Northern Hemisphere. It is clear that School authorities placed the living walls at their locations. It is still not clear how modeling using data from small and partially planted north-facing living walls and a partially vegetated pergola provides valuable information to the larger issues at hand, with urban canyons. Regarding improvements, the Introduction could be shortened. Section 1.1 seems out of place and redundant to information in Section 2. Reporting in K in the text and C in Figures with no translation is difficult to follow/make sense. The pergola has very little vegetation growing on it and seems not relevant to the overall study of UHI. The "very small impact" reported is not yet convincing as valuable to the UHI discussion.

Author Response

Thank you for your comments.

Please see the attachment for details.

Round 3

Reviewer 2 Report

Thank you.